# mmWalk: Towards Multi-modal Multi-view Walking Assistance

**Kedi Ying**[1][*]    **Ruiping Liu**[1][*][†]    **Chongyan Chen**[5]    **Mingzhe Tao**[1]    **Hao Shi**[6]
**Kailun Yang**[3]    **Jiaming Zhang**[1,3,4‡]    **Rainer Stiefelhagen**[1,2]

[1] CV:HCI, KIT    [2] Center for Digital Accessibility and Assistive Technology (ACCESS@KIT)
[3] Hunan University    [4] ETH Zurich    [5] University of Texas at Austin    [6] Zhejiang University

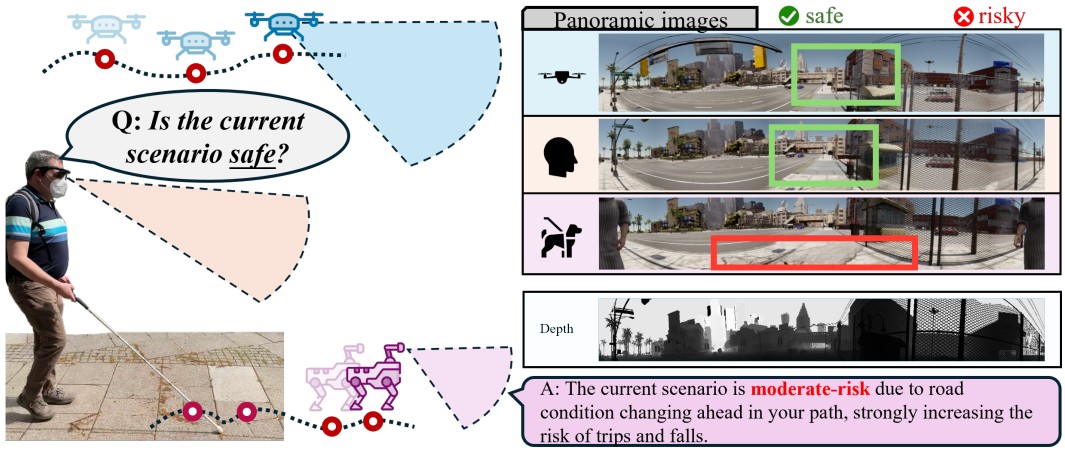

Figure 1: **mmWalk** benchmark with multi-modal (*e.g.*, RGB, depth), multi-view (*i.e.*, drone, walker, guide dog), accessible (corner-case awareness with visual question answering) features.

## Abstract

Walking assistance in extreme or complex environments remains a significant challenge for people with blindness or low vision (BLV), largely due to the lack of a holistic scene understanding. Motivated by the real-world needs of the BLV community, we build **mmWalk**, a simulated multi-modal dataset that integrates multi-view sensor and accessibility-oriented features for outdoor safe navigation. Our dataset comprises 120 manually controlled, scenario-categorized walking trajectories with $62k$ synchronized frames. It contains over $559k$ panoramic images across RGB, depth, and semantic modalities. Furthermore, to emphasize real-world relevance, each trajectory involves outdoor corner cases and accessibility-specific landmarks for BLV users. Additionally, we generate **mmWalkVQA**, a VQA benchmark with over $69k$ visual question-answer triplets across 9 categories tailored for safe and informed walking assistance. We evaluate state-of-the-art Vision-Language Models (VLMs) using zero- and few-shot settings and found they struggle with our risk assessment and navigational tasks. We validate our mmWalk-finetuned model on real-world datasets and show the effectiveness of our dataset for advancing multi-modal walking assistance.
Data: https://doi.org/10.7910/DVN/KKDXDK.
Code: https://github.com/KediYing/mmWalk.

---

[*]Equal contribution. [†] Project lead. [‡] Corresponding author.

39th Conference on Neural Information Processing Systems (NeurIPS 2025) Track on Datasets and Benchmarks.

# 1 Introduction

Blindness and Low Vision (BLV) affect more than 2.2 billion people [1], impacting their ability to travel outdoors and consequently influencing their quality of life and engagement in daily activities. One of the most critical challenges is the clichéd term of outdoor navigation. There are many outdoor navigation aids available, ranging from traditional devices to modern electronic aids to computer vision and AI assistance [2, 3, 4, 5], including a significant proportion of landmark-based navigation systems [6, 7], and a considerable number of notable landmarks for navigating for people with BLV can be found in ATmaps statistics [8]. Despite all of that, the survey [9] indicates that more than 63% of the respondents have experienced at least one incident of injury while navigating in outdoor environments. Furthermore, in [10], it was reported that 7% of individuals with BLV experience at least one fall monthly. A model that prioritizes safety awareness is equally as important as one that ensures answer accuracy. Moreover, there are many scenarios or objects that increase the danger of the current navigation or walking path, including crossing the road, uneven ground, steps, and obstacles on the pavement [11, 12, 13, 14], which makes the term safety very challenging. In this context, an aid that balances hazard awareness and landmark detection is critical and more useful for the BLV community.

Given these challenges, we introduce **mmWalk**, along with **mmWalkVQA**, a novel **m**ulti-view and **m**ulti-modal inclusive **Walk**ing dataset (Figure 1). mmWalk incorporates synchronized frames, referring to a single timestamp, at which multiple panoramic images (from different views) are captured, including all modalities. The frames are collected in the *Carla Simulator* [15] manually, within walker, guide dog, and drone views, capturing rich panoramic pedestrian-egocentric images including RGB, depth, semantic segmentation along with walker's action and inertial measurement unit (IMU) in 120 trajectory path with native action among 7 scenario categories, summing up over $559k$ images. Additionally, we defined 8 corner cases for people with BLV among the aforementioned outdoor dangers and listed 18 valuable navigational landmarks according to ATmaps [8], a European standard platform for summarizing landmarks for tactile maps designed for BLV. The corner cases and landmarks, along with scenario descriptions and weather conditions, are stored in the contextual metadata. Figure 2 gives an example of a parking area scenario trajectory, with a few examples of corner cases and landmarks. In each trajectory, subsets have been further annotated through frame sampling, which were used to generate $69k$ Visual Question-Answering (VQA) for **mmWalkVQA** pairs by GPT-4o [16] in 9 VQA-types of 3 difficulty levels, further enabling extensive benchmarking and a series of experiments with state-of-the-art large language and vision-language models. Section 3.1 describes the collection phase and the construction of the dataset, with a comprehensive presentation and deep analysis of the dataset structure and content.

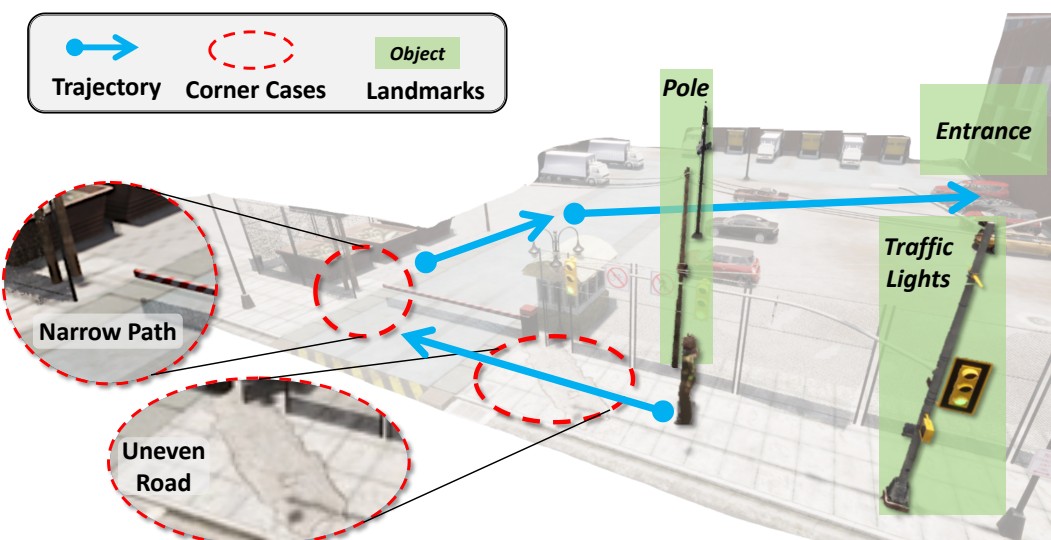

Figure 2: Visualization of a challenging walking scenario of the established mmWalk dataset. While walking in a parking area (the trajectory is the marked blue path), corner cases include *narrow path* and *uneven road*. The navigational landmarks for BLV include *pole*, *traffic light*, and *entrance*.

With mmWalk and mmWalkVQA, we provide a comprehensive analysis of existing open-source Vision-Language Models (VLMs) with multi-image capability, revealing critical limitations in the performance. With the metric of LLM-evaluation, we calculated the normalized score from GPT-4o-mini [16] upon the zero-shot and 3-shot inference results of tested models. Besides, the comparison spans multiple dimensions, including scores across different tasks (VQA types), different scenarios, and different inputs. Furthermore, our analysis demonstrates that even state-of-the-art models struggle with risk assessment and navigational tasks, which provide valuable insights for future model development and optimization, specifically targeted at assistive technologies for the BLV community. We also demonstrate the generalization of a model fine-tuned on our task when deployed for real-world outdoor visual question answering.

In summary, our main contributions are as follows:

- We introduce **mmWalk**, a novel multi-view and multi-modal dataset specifically designed for inclusive walking assistance, encompassing synchronized frame data from walker, guide dog, and drone perspectives with comprehensive modalities (RGB, depth, semantic segmentation), contextual metadata, and a large size (over $559K$ images).

- We provide a scalable pipeline to generate BLV-oriented VQA pairs, offering an accessible and inclusive benchmark (**mmWalkVQA**) for evaluating VLMs in assistive tasks, including scene understanding, pedestrian navigation, and risk assessment for individuals with BLV.

- We analyze the performance of many VLMs on mmWalkVQA, revealing significant limitations in their ability to reason about spatial relationships, identify hazards, and comprehend multi-view scenes from the perspective of BLV users. Cross-evaluation on the real-world dataset proves that VLMs obtain significant benefits by fine-tuning on the established mmWalk dataset.

## 2 Related Work

**Walking and Navigation Assistance.** Multi-view assistance systems represent an important advancement in outdoor navigation. For example, the BLV assistant OpenMPR [17] is a place recognition system utilizing multi-view image data for place matching. The Multi-view Street Scene Perception (MSSP) system [18] uses multiple camera views to enhance the perception of pedestrian paths and obstacles in complex urban environments. In the navigation domain, researchers have investigated how to deploy multi-view sensors [19, 20], particularly drones, to build a multi-view navigation assistant [21, 22, 23]. Despite these technological advances, many navigation systems fail to adequately address safety-critical aspects such as identifying hazardous conditions, uneven surfaces, and temporary obstacles [9, 10]. To overcome this limitation, the proposed mmWalk aims to cover safety-critical factors through its focus on corner cases and accessible landmarks.

**Visual Assistive Datasets.** Numerous visual assistive datasets have been developed for the BLV community, focusing on indoor navigation, object recognition, and text-to-speech conversion for reading assistance. Among these, VQA datasets are particularly prevalent and relevant to our work. For example, the VizWiz dataset [24] contains over 31,000 image-question pairs where visually impaired individuals captured images and asked questions to learn about their surroundings. The recent GuideDog dataset [25] represents a significant step toward egocentric multi-modal data collection for BLV assistance, along with VQA pairs. SideGuide [26] incorporates spatial and depth information with egocentric perspectives specifically for BLV users. More generally, there are pedestrian datasets that are not specifically designed for BLV users but can still be applied to BLV-related scenarios and navigation tasks, such as TBRSD [27], X-World [28], SANPO [29], EGO4D [30], Musohu [31], SpatialLLM [32], and the space-aware instruction tuning dataset [33]. Unlike existing datasets, our mmWalk dataset involves corner cases of walking scenarios that hinder the generalizability of blind assistive systems, providing BLV with accessible landmarks for navigation.

**Corner Cases for BLV.** People who are Blind or with Low Vision (BLV) often face unique safety challenges when navigating outdoor environments. Research on BLV navigation challenges has identified several critical outdoor corner cases. *Road crossing* consistently emerges as one of the most significant concerns, with studies indicating that they are among the highest-risk activities for BLV pedestrians [13, 14, 34, 35]. Additionally, *uneven ground or road* outdoors is considered one of the dangerous challenges [13, 36]. Specifically, [36] specifically mentioned *irregular pavements*, *unknown stairways*, and *roadside potholes* as dangerous factors. Moreover, a study [37] discusses the challenges of dealing with *barriers* on the road, including recognizing objects and getting around

obstacles on the road. [38] points out that obstacles on the road, such as mailboxes and parked motorbikes, can make navigation much harder, creating a more *narrow path* that can be walked through. Another challenge is *finding entrances and exits* to houses, buildings, or underground stations [39, 40, 41]. Maxime *et al.* [42] have made specific reference to the dangers of *obstacles in high positions* such as sagging tree branches and street vendors' awnings. By incorporating these corner cases into the fresh mmWalk dataset, our work aims to enable the development of more comprehensive and safety-oriented navigation assistance systems that can better address the full spectrum of challenges faced by BLV individuals in outdoor mobility.

# 3  Dataset Creation

## 3.1  mmWalk Dataset

Our **mmWalk** dataset was collected using Carla [15], a popular open-source simulator for autonomous driving with customization capabilities and supporting a variety of sensors, which allows us to customize the trajectory and collect an ego pedestrian dataset. mmWalk dataset contains 120 trajectories across of 7 scenario categories and 5 weather conditions (Figure 3). The trajectories have an average of 518 frames. There are three uniquely deployed sensor-groups for each view. With multiple views and multiple modalities including depth and semantic segmentation, mmWalk provides 62,167 frames and over $2.5M$ single images in total, which are collated as a cubemap per frame and converted into an equirectangular panoramic image by the py360convert toolkit [43], summing up 559,503 panoramic images. In addition, we labeled and stored the trajectory metadata which contains trajectory descriptions, occurred BLV corner cases, special landmarks, and the action of the ego pedestrian. More specifics of the dataset collection can be found in Appendix A.

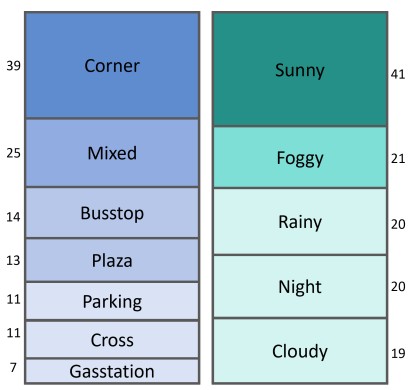

Figure 3: Trajectory numbers of each scenario (left) and weather (right).

**Scenario and Weather.** mmWalk trajectories are strategically distributed across 7 urban scenario categories and 5 weather conditions. Figure 3 shows the number of trajectories in each scenario and weather. The scenario represents a specific type of environment or location characterizing the overall trajectory. In the cases of the 'Corner' scenario, the character has no other special behavioral logic or path goal, but mainly focuses on the corner case, for example, *'bypassing the obstacle on the path'*.

Table 1. Categories and descriptions of the corner cases.

| Corner Case | Description | #Traj. |
|---|---|---|
| Cross road in danger | No traffic light or zebra cross | 16 |
| Cross road | Cross the road generally | 17 |
| Uneven road | Ground condition changes | 40 |
| Barrier | Obstacles in the path blocking the way | 27 |
| Narrow path | Walking through a narrow path | 36 |
| Entrance locating | Find path onto a small entrance | 22 |
| High obstacles | High position obstacles | 11 |
| Deadend | Walking into a deadend | 5 |

**Corner Cases and Landmarks.** As introduced in Section 2, we identify and summarize 8 corner cases critical for BLV users, based on prior literature, and incorporate them into mmWalk. Table 1 provides a succinct and clear description, as well as the number of trajectories containing each corner case category. Note that in the non-corner scenario, depending on our pathway design, one or even more types of corner cases will likewise appear. We can see that critical corner cases such as *uneven road*, *road crossing* (sum up the general crossings and dangerous situational crossings), and *narrow paths* appear very frequently in the dataset. Walking into a *dead-end* road occurs the least, and usually occurs together with *entrance locating*.

For accessibility landmarks, we refer to the list of the most important landmarks identified by blind people, as surveyed in ATmaps [8]. From the list, we then select the 18 items that appear more frequently in the simulator. The specific details of these landmarks are provided in Appendix A.1.

## 3.2 VQA Types and Generation

**VQA Types.** To create **mmWalkVQA**, we designed a total of 9 visual question categories, grouped into three difficult levels: *easy*, *medium*, and *hard*, as shown in Table 2, with detailed categories and distributions.

**VQA Generation.** We randomly sample 7570 frames among all trajectories for mmWalkVQA generation from the mmWalk dataset. We then used GPT-4o to generate the VQA pairs in batches. The overall workflow of VQA generation is shown in Figure 4.

In terms of scene information extraction, we kept the RGB images in all views, adhering to our multi-view concept, while translating the

Table 2. Statistics of VQA types grouped by difficulty levels and categories.

| Difficulty | Category | Description | Count |
|---|---|---|---|
| Easy | E1 | Weather and Action | 8,283 |
| | E2 | Existence | 8,019 |
| | E3 | Counting | 7,586 |
| | E4 | Attribute | 7,570 |
| Medium | M1 | Spatial | 7,670 |
| | M2 | Description | 7,570 |
| | M3 | View Comparison | 7,553 |
| Hard | H1 | Risk Assessment | 7,570 |
| | H2 | Navigational Landmarks | 7,570 |

semantic segmentation and depth images into a list of strings describing object positions in a **BLV-friendly clockwise manner** [44, 45, 46]. Figure 4 illustrates the clockwise spatial descriptions from the ego perspective, where the blue nodes represent the semantic segments. The string lists were merged together with the contextual metadata. The system and instruction prompt are detailed in the Appendix A.2. Additionally, we manually crafted 1∼3 example pairs for each VQA category. Those are fed into ChatGPT-4o to generate VQA pairs.

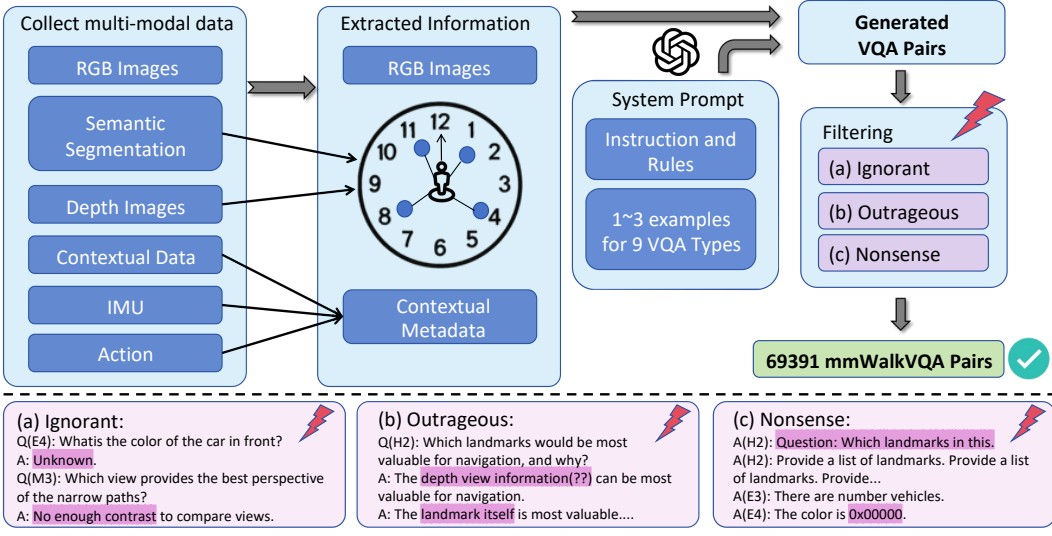

Figure 4: Workflow of VQA triplets generation.

**VQA Filtration and Validation.** To ensure high data quality, we first applied automatic filtering to remove low-quality VQA pairs by identifying manually defined keywords in questions and answers, such as *Unknown, not possible, etc.*, as illustrated at the bottom of Figure 4. We iteratively refined the list of keywords until the filtering results were satisfactory. Additionally, manual corrections were applied to question-answer pairs when a quick fix was feasible. We also randomly sampled 270 VQA pairs, and the authors manually examined them for quality. The manual exam result can be found in Table 3, reporting average scores from the manual evaluation of 270 sampled pairs across four criteria, where 5 is the perfect score and 1 is the lowest score. In particular, *Answer Actionability* refers to whether the response provides users with clear, actionable guidance for decision-making.

This process resulted in a final set of 69,391 VQA triplets under 3 difficulty levels and 9 VQA types. We then split the dataset based on trajectory to ensure that the validation and testing sets do not contain similar scenes. Table 3 provides an overview of the training, validation, and test splits of the dataset.

Table 3. The data statistics and data splits of the established mmWalkVQA dataset.

| Difficulty Level | Data Split | | | Average Word Length | | Quality Score | | | |
|---|---|---|---|---|---|---|---|---|---|
| | Train | Val | Test | Question | Answer | Question Quality | Answer Correctness | Answer Actionability | Answer Fluency |
| Easy | 25140 | 3214 | 3166 | 7.44 | 3.85 | 4.53 | 4.70 | 4.66 | 4.80 |
| Medium | 18201 | 2318 | 2249 | 8.69 | 22.74 | 4.71 | 4.87 | 4.84 | 4.91 |
| Hard | 12079 | 1537 | 1487 | 9.62 | 30.67 | 4.55 | 4.62 | 4.43 | 4.81 |

**Dataset Analysis.** We compare mmWalkVQA and five related datasets and benchmarks in Table 4. Overall, mmWalkVQA features the largest size of VQA pairs and the most diverse Q types. Uniquely, it incorporates multi-view, panoramic, sequential trajectory, and egocentric images, yielding contextual richness and spatial accuracy of the data. These characteristics not only broaden the range of applicable scenarios but also enable the design of specialized VQA tasks tailored to the needs of the Blind and Low-Vision (BLV) community. We also conducted a statistical analysis of the average word length, as presented in Table 3, and found that, as expected, longer questions and answers are associated with higher levels of difficulty.

Table 4. Comparison of datasets and benchmarks across multiple features with △ indicates partially satisfied. MV: Multi-View. PA: Panoramic images. SD: Spatial or Depth. EC: Egocentric. BLV: BLV Guidance/Assistance. Seq: Sequential Trajectories. #VQA: Number of VQA Pairs. #VQA Type: Number of VQA types. FT: finetuned Evaluation.

| Dataset | Year | MV | PA | SD | EC | BLV | Seq | #VQA | #VQA Type | FT |
|---|---|---|---|---|---|---|---|---|---|---|
| Vizwiz [24] | CVPR'18 | × | × | × | ✓ | ✓ | × | 31,173 | 4 | ✓ |
| SideGuide [26] | IROS'20 | × | × | ✓ | ✓ | ✓ | × | × | × | × |
| SANPO [29] | WACV'25 | ✓ | × | ✓ | ✓ | △ | ✓ | × | × | × |
| EgoTextVQA [47] | CVPR'25 | × | × | × | ✓ | × | ✓ | 7,064 | 5 | × |
| GUIDEDOG [25] | arXiv'2503 | × | × | ✓ | △ | ✓ | ✓ | 818 | 2 | ✓ |
| **mmWalkVQA (ours)** | | ✓ | ✓ | ✓ | ✓ | ✓ | ✓ | 69,391 | 9 | ✓ |

# 4 Model Benchmarking

## 4.1 Baseline Models

For benchmarking, we selected a number of open-source, multi-image input-enabled visual language models with language model sizes of 7∼8B, including **LLaVA One-Vision** [48], **LLaVA Next** [49], **Qwen2VL** [50], **InternVL2** [51], **Janus-Pro** [52], and **Chameleon** [53].

## 4.2 Evaluation Metric

In our experiments, following *LLM as-a-judge* [54, 55], we use GPT-4o-mini [16] to evaluate the similarity and correctness of the output answers generated by every model and the ground truth answers generated by GPT-4o, which are presupposed in our annotation and mentioned in Section 3.1. We designed a unified custom scoring prompt (detailed in Appendix A.2) for evaluation of all models, which is input to GPT-4o-mini along with the questions, generated answers, and ground truth answers. The metric is a scaled score (1∼5, with 5 being the highest score). In order to visualize the differences in model performance, we normalized all scores to 0∼100 in all tables of experimental results. The data represented is the normalized scores $S_{normalized}$, which are given by the formula:

$$S_{normalized} = \frac{1}{N_{samples}} \sum_{i=1}^{N_{samples}} \frac{s_i - 1}{4} \times 100\%,$$

where $s_i$ represents the original score of the $i$-th sample and $N_{samples}$ represents the number of all samples. Note that the data presented in all tables are accurate to two decimal places ($.2f$), which could lead to minor errors ($\leq 0.1$) in calculated average scores.

## 4.3 Overall Results

Table 5 presents the performance of all models across each VQA category under both the zero-shot and 3-shot setups, accompanied by a composite average score.

Overall, we found that all the models struggle on this task for all VQA categories with each model exhibiting strengths and weaknesses across different categories. These results indicate that the mmWalkVQA task presents substantial challenges. They also reveal inconsistencies in the capabilities of current open-source models. The result shows that InternVL2 performs the best among the zero-shot models, slightly outperforming LLaVA-Next and Qwen2VL. InternVL2 has a superior performance in the M3 category over other models, and a stable performance in all other categories. We thus chose InternVL2-8B for fine-tuning and evaluated 3-shot performance for the remaining models. Among the three-shot models, Qwen2VL-7B or Janus-Pro-7B consistently achieved the highest scores across all VQA categories. However, LLaVA-Next consistently ranked in the top three in most categories, resulting in the highest average score.

Table 5. Results on mmWalkVQA over all VQA categories, taking RGB panoramic image of all three views as input. The **best zero-shot model** and the **best 3-shot model** for each VQA category and average score have been highlighted. The last column shows the improvement in scores at the 3-shot or finetuned model compared to the same zero-shot model.

| Setting and Model | E1 | E2 | E3 | E4 | M1 | M2 | M3 | H1 | H2 | Average | Improved |
|---|---|---|---|---|---|---|---|---|---|---|---|
| *Zero-shot* | | | | | | | | | | | |
| Chameleon-7B | 25.78 | 20.64 | 16.01 | 15.39 | 7.45 | 19.85 | 29.17 | 28.19 | 27.36 | 21.14 | |
| Janus-Pro-7B | 12.41 | **46.13** | **33.40** | 50.65 | 21.59 | **54.85** | 31.01 | **54.30** | **36.07** | 37.82 | |
| Qwen2VL-7B-Instruct | **84.01** | 40.79 | 24.39 | 52.75 | 22.47 | 50.10 | 14.95 | 11.42 | 32.97 | 37.91 | |
| LLaVA-NEXT-7B | 81.58 | 43.45 | 17.59 | **58.12** | **24.71** | 42.19 | 28.98 | 28.47 | 33.51 | 39.84 | |
| LLaVA-Onevision-7B | 54.21 | 37.93 | 12.45 | 29.11 | 16.21 | 19.11 | 5.07 | 9.68 | 13.12 | 21.87 | |
| InternVL2-8B | 77.56 | 42.66 | 31.22 | 53.12 | 17.03 | 46.51 | **53.26** | 14.92 | 35.94 | **41.35** | |
| *3-shot* | | | | | | | | | | | |
| Chameleon-7B | 27.91 | 22.67 | 15.46 | 16.64 | 13.32 | 8.65 | 33.01 | 27.89 | 36.84 | 22.48 | 1.34 |
| Janus-Pro-7B | 11.78 | **45.36** | **33.61** | 50.92 | **21.36** | **55.08** | 31.38 | **54.33** | 35.97 | 37.75 | -0.07 |
| Qwen2VL-7B-Instruct | **84.01** | 40.63 | 24.43 | **55.26** | 15.72 | 45.83 | **45.33** | 13.88 | **45.91** | 41.89 | 3.98 |
| LLaVA-NEXT-7B | 83.73 | 41.81 | 17.01 | 53.29 | 13.94 | 46.33 | 40.39 | 47.58 | 43.81 | **43.71** | 3.87 |
| LLaVA-Onevision-7B | 54.21 | 38.02 | 12.35 | 36.07 | 13.32 | 27.32 | 22.88 | 32.46 | 40.31 | 31.21 | 9.34 |
| *finetuned* | | | | | | | | | | | |
| **InternVL2-8B** | **94.15** | **50.86** | **35.84** | **67.12** | **28.05** | **60.33** | **51.18** | **53.71** | **50.37** | **55.21** | **13.86** |

Note: For the LLaVA family, the size of the language models we use remains 7-8B, with LLaVA-Onevision-qwen2-7B and LLaVA-Next-v1.6-mistral-7B used in the table above.

Among all tasks, M1 (Spatial) proved to be the most challenging category both for the zero-shot models and the 3-shot models, as M1 required a comprehensive spatial understanding, combining all the input modalities.

Across prompt settings, most models demonstrated improved performance under 3-shot learning compared to zero-shot. Among them, LLaVA-Onevision gained the largest improvement with more than $9.34\%$ improvement compared to zero-shot. In 0-shot cases, LLaVA-Onevision answered long texts in medium and hard questions with less relevance to key features, resulting in its low scores in M1-H2. With three examples provided, LLaVA-Onevision was better able to focus on the relevant aspects in its responses. Janus's score decreased slightly, likely because it tended to mimic the example answers rather than reasoning from the input images while facing medium-level questions in our tasks.

When comparing model (*i.e.*, InternVL2) with and without fine-tuning, we found that it improved its score by $13.86\%$, which proves the effectiveness of training on our proposed mmWalkVQA dataset.

## 4.4 Fine-Grained Analysis

**Analysis of Different Walking Scenarios.** Table 6 shows results of models across different Scenario categories. For each model, the highest-scoring scenario is highlighted in the table. The tasks associated with large open area (*e.g.*, Plaza, Parking) generally yielded higher scores among many models. In contrast, scenarios such as Corner, Busstop, and Gasstation, being more intricate and requiring finer-grained understanding, led to lower performance.

Table 6. Results on mmWalkVQA over walking scenarios, taking RGB panoramic image of all three views as input. The **scenario-wise best performance** of each model (each row) has been highlighted.

| Setting and Model | Corner | Cross | Busstop | Mixed | Plaza | Parking | Gasstation |
|---|---|---|---|---|---|---|---|
| *Zero-shot* | | | | | | | |
| Chameleon-7B | 18.66 | **24.36** | 18.59 | 21.68 | 22.60 | 21.71 | 20.74 |
| Janus-Pro-7B | 36.50 | 31.29 | 32.64 | 42.37 | **45.35** | 29.77 | 35.84 |
| Qwen2VL-7B-Instruct | 33.81 | 34.57 | 34.52 | 38.76 | **45.14** | 39.91 | 34.09 |
| LLaVA-NEXT-7B | 37.65 | **41.12** | 37.11 | 40.17 | 38.95 | 37.46 | 37.26 |
| LLaVA-Onevision2-7B | 21.36 | 26.40 | 15.48 | 24.81 | 26.54 | **28.55** | 12.55 |
| InternVL2-8B | 38.26 | 41.95 | 41.01 | 42.37 | **45.08** | 42.36 | 40.77 |
| *3-shot* | | | | | | | |
| Chameleon-7B | 22.28 | 22.81 | 21.22 | 23.81 | 23.45 | **23.65** | 19.27 |
| Janus-Pro-7B | 36.23 | 31.47 | 32.97 | **44.89** | 42.07 | 30.07 | 35.75 |
| Qwen2VL-7B-Instruct | 39.31 | 38.40 | 37.99 | 42.93 | **48.36** | 44.01 | 38.12 |
| LLaVA-NEXT-7B | **47.65** | 45.02 | 42.54 | 42.04 | 42.03 | 45.11 | 43.84 |
| LLaVA-Onevision2-7B | 26.98 | 35.69 | 24.02 | **38.70** | 32.98 | 33.61 | 24.24 |
| *finetuned* | | | | | | | |
| InternVL2-8B | 53.67 | 51.94 | 54.25 | **59.49** | 56.87 | 50.17 | 56.58 |

Table 7. Results of different input views. Full stands for full view with walker, dog, and drone. Scores **better than full view** have been highlighted, otherwise worse.

| Models and Inputs | E1 | E2 | E3 | E4 | M1 | M2 | M3 | H1 | H2 | Average without M3 | Δ to Full |
|---|---|---|---|---|---|---|---|---|---|---|---|
| *InternVL2-8B (finetuned)* | | | | | | | | | | | |
| Full | 94.15 | 50.86 | 35.84 | 67.12 | 28.05 | 60.33 | / | 53.71 | 50.37 | 55.05 | / |
| Walker | 93.12 | 48.34 | **38.49** | 62.70 | 27.89 | 59.02 | / | 49.63 | 46.33 | 53.19 | **-1.86** |
| Walker+Dog | 93.40 | 50.86 | **37.35** | **62.98** | **28.71** | 59.93 | / | 51.04 | 46.74 | 53.85 | **-1.20** |
| Walker+Drone | 94.10 | 50.15 | 35.68 | **67.22** | **28.61** | **60.80** | / | 49.19 | 46.91 | 54.08 | **-0.97** |
| *LLaVA-NEXT-7B (zero-shot)* | | | | | | | | | | | |
| Full | 81.58 | 43.45 | 17.59 | 58.12 | 24.71 | 42.19 | / | 28.47 | 33.51 | 41.20 | / |
| Walker | 74.86 | 42.54 | 15.23 | 51.29 | **25.07** | **43.61** | / | 27.72 | **36.47** | 39.59 | **-1.61** |
| Walker+Dog | 49.31 | 42.72 | **19.31** | 54.31 | 23.98 | **42.83** | / | 22.83 | **35.26** | 36.06 | **-5.14** |
| Walker+Drone | 70.01 | **43.52** | **17.91** | 54.21 | **24.80** | 41.89 | / | 22.11 | **34.25** | 38.58 | **-2.62** |
| *Qwen2VL-7B (zero-shot)* | | | | | | | | | | | |
| Full | 84.01 | 40.79 | 24.39 | 52.75 | 22.47 | 50.10 | / | 11.42 | 32.97 | 39.86 | / |
| Walker | 78.23 | **41.62** | 23.90 | 48.34 | **24.02** | **51.75** | / | **12.20** | **33.48** | 39.19 | **-0.67** |
| Walker+Dog | 78.63 | 39.50 | 22.79 | 49.52 | 21.56 | **52.36** | / | 11.36 | 32.74 | 38.55 | **-1.31** |
| Walker+Drone | 77.82 | **41.41** | **26.27** | 50.82 | **23.59** | **50.40** | / | 11.20 | 32.44 | 39.36 | **-0.50** |

**Analysis of Multi/Double/Single view Inputs.** In real-world scenarios, BLV users often navigate independently, accompanied by a guide dog, or assisted by a drone, while full visual coverage may not always be accessible. Therefore, we conducted experiments using three input configurations: walker only, walker plus guide dog view, and walker plus drone view. The models we benchmarked are the top-3 models from our base experiment. Results are shown in Table 7. As expected, full view generally outperforms single or dual views across all models on average. However, when examining specific tasks, we observed instances where single or dual views performed better, as highlighted in green, with an example shown in Figure 5, where the question M1 can be answered easily by any views. Notably, for tasks like H1 (risk assessment), the drone view proved to be the least effective. This may be attributed to the drone's elevated perspective, which can miss ground-level risks such as uneven surfaces or narrow pathways. An illustrative example is shown in Figure 1, where the risk is only captured by the dog view.

**Reliability of LLM-as-a-judge** While LLM-based evaluation has become a widely adopted practice in recent works, the debate over whether LLM-as-a-judge is reliable remains ongoing. To further strengthen the reliability of the outcomes, we conducted an extra human rating study to directly quantify its consistency. Specifically, we randomly sampled 10 percent of VQA pairs from five model outputs, including Finetuned InternVL2-8B, Qwen2 in zero-shot and 3-shot settings, LLaVA-OneVision in the 3-shot setting, and LLaVA-Next in zero-shot. In total, 3,575 VQA samples across all QA categories were rated by human annotators. We then calculated **Spearman's** $\rho$ [56] between the human ratings and GPT-4o evaluation scores.

Our results show that the finetuned model (InternVL2-8B) achieved a **Spearman's** $\rho$ of 0.924, indicating very high agreement between human and GPT scores. Across all five models, the average **Spearman's** $\rho$ reached 0.864, which demonstrates the overall reliability and consistency of our automatic evaluation pipeline. This fine-grained analysis confirms that our adoption of LLM-as-a-judge provides robust and scalable evaluation while maintaining high correlation with human judgment.

**Qualitative Examples.** In Figure 5, we show three output examples of our finetuned *InternVL2-8B* with full multi-view input. These examples show that our model is capable of providing reasonable answers for medium and hard VQA categories.

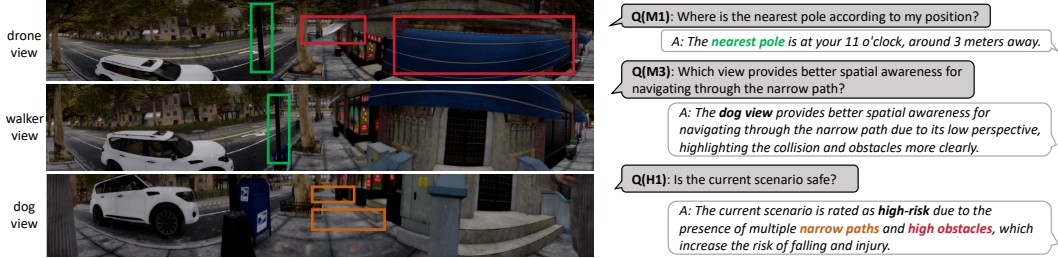

Figure 5: Qualitative examples of output from finetuned InternVL2-8B.

## 4.5 Generalization on Real-World Dataset

To present the generalization of our dataset, we evaluated the model finetuned on our dataset on the related realistic dataset, EgoTextVQA [47], which contains outdoor scene VQs.

Following the official scoring methodology from EgoTextVQA, we compared the performance of mmWalk finetuned *InternVL2-8B* with the originally reported results. mmWalk finetuned the model with multi-image inputs, so we chose to use frames as input. The official EgoTextVQA paper published the *InternVL2-8B* scores, including the score of each category with video input, and overall scores with frame input (first and second rows of the Table 8). Our model improves in almost all EgoTextVQA categories as well as in the overall score, as highlighted in the table.

Table 8. Results of cross dataset evaluation following EgoTextVQA [47, 57]. The score is rated by GPT-4o with accuracy/score (from 1 to 5, the higher the better). The **better performance** compared with EgoTextVQA InternVL on frame input and video input has been highlighted. EgoTextVQA did not publish scores for each category for frame input; consequently, undisclosed scores are denoted as n.a. in the table.

| Model | Input | Location | Direction | Description | Intention Reasoning | Others | Overall |
|---|---|---|---|---|---|---|---|
| EgoTextVQA InternVL2-8B | Video | 15.8/1.4 | 21.9/1.7 | 14.8/1.0 | 14.5/1.2 | 13.6/1.3 | 16.4/1.3 |
| EgoTextVQA InternVL2-8B | Frame | *n.a.* | *n.a.* | *n.a.* | *n.a.* | *n.a.* | 18.5/1.4 |
| **mmWalk-finetuned InternVL2-8B** | Frame | 11.82/1.59 | **22.58/2.05** | **29.70/2.11** | **23.12/1.95** | **27.22/2.24** | **21.55/1.92** |

## 5 Conclusion and Discussion

We present mmWalk, a multi-modal multi-view dataset for benchmarking walking assistance for individuals with Blindness or Low Vision (BLV). Our work addresses a critical gap in existing datasets by combining comprehensive multi-modal data with multiple viewpoints and a deliberate focus on corner cases and navigational landmarks specific to BLV users. mmWalk uniquely features panoramic views, explicit annotation of BLV-relevant corner cases, and special navigational landmarks identified through ATmaps statistics. The accompanying mmWalkVQA benchmark enabled systematic evaluation of VLMs on BLV-relevant tasks, revealing significant performance gaps in state-of-the-art models, particularly in complex tasks like risk assessment and navigational landmark searching.

# 6 Broader impacts, limitations, and future work.

We expect our work to directly benefit BLV users by enhancing daily walking and navigation assistance and hope it can raise awareness among today's VLM developers for more inclusive models, thus having positive societal impacts. Additionally, it can support broader research communities in computer vision, including VQA, image captioning, pedestrian navigation, autonomous driving, robotics, and embodied AI.

While promising, there are areas for improvement. Firstly, to comply with GDPR regulations, we collected the mmWalk dataset exclusively within a simulated environment, ensuring that no personal or sensitive information from real-world scenarios was captured, thereby maintaining data privacy and security. We also demonstrate that our dataset generalizes well to outdoor real-world VQA settings. However, a potential limitation is the risk of biased model behavior, as the training data are synthetic and may not fully capture the diversity of real-world BLV experiences, thus introducing potential negative societal impacts. Future work can address this by collecting data in real-world settings and easily adapting our proposed VQA generalization pipelines to mimic our tasks. Secondly, although using LLM-as-a-judge for answer evaluation has proven effective, it can also introduce biases. Future work should investigate these biases and develop more stable evaluation metrics. Lastly, while mmWalk leverages multi-view and multi-modal features, some modalities, such as IMU data, sequential frames, and semantic labels, can be further explored. Expanding mmWalk to include these features more comprehensively could also open new avenues for tasks beyond VQA, such as image captioning and embodied AI training.

# 7 Acknowledgment

This work was supported in part by the Ministry of Science, Research and the Arts of Baden-Württemberg (MWK) through the Cooperative Graduate School Accessibility through AI-based Assistive Technology (KATE) under Grant BW6-03, in part by funding from the pilot program Core-Informatics of the Helmholtz Association (HGF), in part by Karlsruhe House of Young Scientists (KHYS), and in part by the Helmholtz Association Initiative and Networking Fund on the HAICORE@KIT and HOREKA@KIT partition. This project is also supported in part by the National Natural Science Foundation of China under Grant No. 62473139, in part by the Hunan Provincial Research and Development Project (Grant No. 2025QK3019), and in part by the Open Research Project of the State Key Laboratory of Industrial Control Technology, China (Grant No. ICT2025B20).

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

# A mmWalk Dataset Details

The mmWalk manual dataset collection is divided into three main parts, design, collection, and annotation. All code scripts are original and were collected on personal computers, and it is still under discussion whether the dataset collection scripts will be made public or not. Note that the collection requires Python 3.7 and a certain PC configuration to run Carla Simulator and pygame [58]. Please refer to the official Carla Simulator webpage [15] for detailed requirements. To ensure full compliance with GDPR regulations, we collected the mmWalk dataset exclusively within a simulated environment. This approach eliminates the risk of capturing personal or sensitive information from real-world scenarios, thereby upholding data privacy and security standards.

## A.1 mmWalk

**Design.** After running the Carla software, we can access a Carla's window as shown in Figure 6 in the perspective of *Spectator* under the player's control. We used a total of 9 maps except these when designing the paths, mainly referring to the scenario and corner case mentioned in the main topic of the article, which are described in more detail in Tables 9 and 10.

**Collection.** Depending on the requirements of our completed trajectory design, we moved the *spectator* to the start point of the trajectory and run our collection script that integrates the selection of the weather settings in Figure 7 and the panoramic image transformation [43]. Once running, a new pygame window will appear that allows us to control the pedestrian's movement from the walker's first-person perspective, capturing keyboard inputs as action data and capturing original images in 4 directions for walker and dog view(left, front, right, back) and 6 directions for drone view(additional up and down) at a 2-second frame rate, meanwhile, dynamic moving hazards such as moving vehicles on the road will automatically spawn at random location nearby the ego pedestrain, controlled by the AI in the simulator with preset action and logic. The converted panoramic images are saved in a preset dictionary format as Figure 8.

**Annotation.** The metadata document can be found in the link to the dataset. The metadata information is all manually annotated, which includes the trajectory name, id, description, occurrence of corner cases, occurrence of important landmarks (Table 11), and the number of frames, and Table 12 provides the style of the metadata information. Of the 120 trajectories, 103 contained at least one type of corner cases, and 114 contained at least one type of landmarks.

## A.2 mmWalkVQA

Table 13 provides details of the VQA categories and Table 14 lists examples of each category given to GPT-4o [16] along with the prompt in Figure 9. Figure 10 shows some examples of what is handled in the filtering job. Figure 11 gives good examples of generated pairs for hard-level VQA categories. Note that all the inputs for generating the VQA pairs are 3 RGB images in different viewpoints and a spatial information matrix, here we only provide a certain image (or a certain part of it) for brevity.

# B Experiment Details

## B.1 Models

Table 15 shows the list of working environment resources and Table 12 shows the parameters of LoRA-finetune, the merged finetuned model and associated parameters are publicly available. Our finetune work took 10 hours per 2 Epochs, and for the inference in other experiments, the time consumed varied from 3~6 hours with the performance of the model. Please check our submitted codes for more job-related information.

## B.2 Evaluation

Figure 13 illustrates the prompt used to score the model evaluation on a scale of 1~5. After running the scoring, the script automatically gives a normalized score for each item based on the VQA category and the Scenario category, respectively, as well as an overall score with values accurate to .2f.

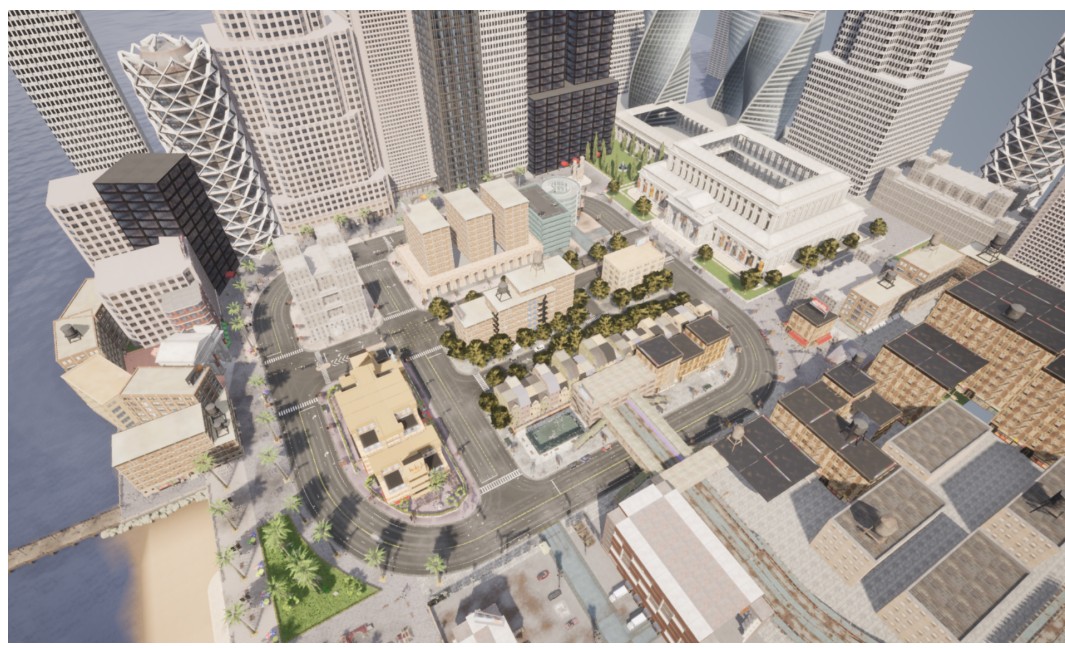

Figure 6: Overview of Carla map in Spectator perspective.

Table 9. Scenario and Detailed Description

| Scenario | Description |
|---|---|
| Busstop | Trajectory starts near a bus stop, or the target location is heading to a bus stop. |
| Gasstation | The entire trajectory uses the gas station or objects within the gas station (*e.g.*, parked cars, shops) as the target location or starting location. |
| Cross | The trajectory is based on the theme of crossing a road or multiple crossings. |
| Parking | Trajectories that start or end with a core theme of car parks, with vehicles inside the car parks or entrances and exits to the corresponding buildings. |
| Plaza | The theme of open squares and plaza, walking through a large square throughout or looking for specific objects in the square (*e.g.* benches, restaurants). |
| Mixed | Trajectories that mix two or more scenarios of all the above. |
| Corner | Trajectories are designed purely on the basis of encountering and solving the BLV Corner Cases. |

Table 10. Corner Case with detailed description

| Corner Case | Detail Description |
|---|---|
| Entrance Locating | Searching correct path onto a small entrance of a house or a building. |
| High obstacles | High position obstacles may hurt head, face and influent the sensors |
| Deadend | Walking into a deadend, including turning around and walking back. |
| Uneven Road | Ground condition changes consisting of changes in topography, changes in ground materials, standing water due to rain, broken glass bottle residue, going up and down stairs, cross a bridge *etc.* |
| Cross the Road in danger | Cross the road without traffic light or without pedestrian cross. |
| Cross the Road | Cross the road generally. |
| Barrier | Obstacles in the path blocking the way, making it necessary to make a diversion, including vehicles, moving boxes, bikes or motorcycles occupying the path, terrain, large bushes, etc. |
| Narrow Path | Walking through a narrow path that may be created by a natural scene (trees, plants, bushes) or by vehicles, buildings, traffic lights, utility poles, etc. |

```
Sunny = {
  cloudiness=0, precipitation=0, precipitation_deposits=0,
  wind_intensity=30, wetness=0,
  sun_altitude_angle=75.0 }
Rainy = {
  cloudiness=80.0, precipitation=60.0, precipitation_deposits=40.0,
  wind_intensity=40.0, wetness=60.0,
  sun_altitude_angle=45.0 }
Foggy = {
  cloudiness=50.0, precipitation=0.0, precipitation_deposits=0.0,
  wind_intensity=30.0, wetness=0,
  sun_altitude_angle=60.0,
  fog_density=65.0, fog_distance=10.0, fog_falloff=1.0 }
Cloudy = {
  cloudiness=80.0, precipitation=0.0, precipitation_deposits=0.0,
  wind_intensity=50.0, wetness=0,
  sun_altitude_angle=65.0,
  fog_density=0.0 }
Night = {
  cloudiness=20.0, precipitation=0.0, precipitation_deposits=0.0,
  wind_intensity=15.0, wetness=0,
  sun_altitude_angle=−30.0,
  sun_azimuth_angle=270.0 }
```

Figure 7: Weather configuration.

```
../Dataset
    /Busstop01
        /dog
            /rgb
                000001.png
                000002.png
                ......
            /semantic
                ......
            /depth
                ......
        /walker
            ... # Same as /dog Folder
        /drone
            ... # Same as /dog Folder
        /imu
            000001.txt
            000002.txt
            ......
        /action
            000001.txt
            000002.txt
            ......
    /Busstop02
    ......
```

Figure 8: Dataset Dictionary.

Table 11. Landmark List

| id | content | comment |
|---|---|---|
| 1 | Traffic Lights | |
| 2 | Bus Stop | |
| 3 | Entrance or Exit | Take in count only if the exit or entrance are preset as spawn or goal |
| 4 | Stairs | |
| 5 | Square | including foundation, sightseeing square, square front of the church *etc.* |
| 6 | Pedestrian Cross | annotated only if the pedestrian cross the road |
| 7 | Garbage Bin | small garbage bin |
| 8 | Dumpster | large garbage dumpster |
| 9 | Gate | large gate of parking area, construction site *etc.* |
| 10 | Bench | |
| 11 | Motocycle/Bycicle | |
| 12 | Poles | including streetlights and electric poles |
| 13 | Postbox/Mailbox | |
| 14 | Map Board | |
| 15 | High voltage box | marked as dangerous for risk assessment |
| 16 | Manhole Cover | marked as dangerous for risk assessment |
| 17 | Roadside Stall | including Food stalls, newsstands, vending machines |
| 18 | Money ATM | |

Table 12. Example Metadata of 4 different trajectories. / indicates no special landmark annotated. The same number in ID indicates the same path of these trajectories, yet opposite direction or different weather for comparison.

| Scenario | ID | Trajectory Description | Appeared Corner Case | Landmark id | Frame | Weather |
|---|---|---|---|---|---|---|
| Busstop | 14 | From bus station to home at night, walking alongside a narrow sidewalk | Narrow Path, Entrance Locating | 2,3,7,12,15 | 551 | Night |
| Corner | 20A | Walking passby a Barrier on sidewalk, go through multiple narrow path and walk into a yard through uneven road. (direction A) | Narrow Path, Barrier, Uneven Road | / | 296 | Foggy |
| Corner | 20B | Walking from a yard through uneven road, go through multiple narrow path and walk passby a Barrier on sidewalk. (direction B) | Narrow Path, Barrier, Uneven Road | / | 271 | Foggy |
| Mixed | 20 | Climb up stairs, cross 2 road continuously in one intersection to reach the bus stop | Uneven Road, Cross the road | 1,2,4,6,12 | 1249 | Sunny |

Table 13. Detailed Information of VQA Types

| ID | Content | Type |
|---|---|---|
| E1 | Weather & Action | query action and weather |
| E2 | Existence | query existence of one landmark, corner case or objects |
| E3 | Counting | counting landmarks or objects |
| E4 | Attribute | query attributes based on rgb-images |
| M1 | Spatial | query relative spatial information based on rgb image and translated spatial description |
| M2 | Description | full description of current scene |
| M3 | View Comparison | compare different views, in concern corner cases or landmarks |
| H1 | Risk Assessment | assess the risk level of current situation |
| H2 | Navigational Landmarks | evaluate landmarks about navigational value |

SYSTEM_MESSAGE = "Generate 15 QA pairs based on multi−view scene graph and json file represented. A FRAME represents one frame of one trajectory, which contains one json file that describes this certain frame and path of 3 RGB images under three different views(dog, walker and drone), along with text descriptions of transformed semantic and depth information. Generate Question Answer pairs for each FRAME which should cover all 9 QA Types.(At least 1 pair for each type) The Question Answer pairs must follow the instruction and rules below, taking the given sample as example."

INSTRUCTION_CONTENT = "
Rules:
1.Only describe clear information in the images − do not fabricate or invent in the answers.
2.Base all answers only on what is actually visible in the provided rgb images and stated in the json data. Do not make assumptions or invent details.
3.All Position information must be described in clockwise manner.(Instead of left/right, describe exact clockwise location such as 'your 3 o'clock')
Instructions:
Easy Level QA: QA pairs that query the basic information in the json file or single image, the answer can be completely verified by the ground truth. Consider the questioner is the pedestrain in the first−person perspective of every scene, use "my surrounding" or current environment instead of "scene" in question.
−Type E1− General Query: the simple query questions about single feature(weather, action)), the answer should be concise and in several words or at most one sentence.
−Type E2− Existence: Query existence of specific objects or corner cases or landmarks in the current rgb image, the answer should be Yes or No with at most one sentence for extension, don't mention views in question and answers. '
−Type E3− Counting: Inquire the exact number of features, using only walker view, don't mention views in question and answers.
−Type E4− Attribute: simple query questions based on the RGB Image(color,shape,size,texture), the answer should be at most one sentence.
Middle Level QA: QA pairs that contain multiple views or multiple images in concern, the answer stems mainly from the combined ground truth feature information. The answer can be partially verified. Consider the questioner as an analyst instead of the pedestrain.
−Type M1− Spatial: Query about the spatial information (distance,position) between multiple objects.
−Type M2− Description: Describing the scene with all(or many) features depending on the question. Answers must be completely based on the information in multiple features and be at most 3 sentences, depending on the complexity of the scene, rely more on depth information.
−Type M3− Comparison:Compare different three views(dog/walker/drone), answers must be completely relevant to the enquiry and give more detailed reasons only based on ground truth. Answers must be at most 2 sentences, refer the appeared corner case in answer in concrete.
Hard Level QA: QA pairs that based on multiple features and multiple frames but require further merging, processing, analyzing and expanding. The question is on the abstract and summarizing perspective. The answer stems mainly from the processed ground truth information and image, refer the appeared corner case if there is any. The answer must be at most 3 sentences long.
−Type H1− Risk Assessment: Calculate the Risk of the scene based the information. The answer should give the final score in low/moderate/high risk based on image, concern weather, corner cases and risky landmarks and brief explanation in few sentences.
−Type H2− Landmarks and Navigation: Evaluate landmarks in the surroundings based on navigational value, and make an overall evaluation based on the size, distance, and number of landmarks, add concrete but concise reason based on image, not only saying that landmark has fixed position."

IMAGE_DESCRIPTION = "These images show the current scene from three camera positions ( walker at eye level, drone from above, dog from low position). The RGB images show actual colors. Instead of depth and semantic images, we provide text descriptions of objects with their clock positions and approximate distances."

Figure 9: VQA generation prompt.

Table 14. Given examples of each VQA category

| ID | Q | A |
|---|---|---|
| E1-1 | What is the weather now? | Rainy. |
| E1-2 | What is the current action? | Turning Left. |
| E2-1 | Is there any benches in the scene? | No. |
| E2-2 | Does the scene contain any corner cases for blind? | Yes, the scene contains multiple barriers and one narrow path. |
| E3 | How many vehicles are there in the parking area? | 3 |
| E4-1 | What is the color of the car in gas station? | Black |
| E4-2 | How large is the Barrier in front of me? | There are some large cardboard boxes stacked on top of each other on the pavement, 1 meter high and 1 meter wide. |
| M1-1 | What is the relative position of the sidewalk to the bus stop? | The bus stop is located on only one sidewalk, you are currently standing on the side with bus stop. |
| M1-2 | How far is the intersection? | The intersection is around 20 meters far from you. |
| M1-3 | Where is the bus stop according to my position? | The bus stop is 12 meters far at your 4 o'clock. |
| M2 | Can you describe the scene | You are currently standing in the middle of a square, there are few roadside stall nearby. There are multiple tables on your 2 o'clock. There is a bus stop 16 meters far at your 7 o'clock. |
| M3 | Which view gives more information about the dangerous environment? | You are about to cross multiple narrow paths, the obstacles are clearer in the 'dog' view. Whereas the drone view is partially obscured by the foliage as the high obstacle. The dog view therefore provides the most information to ensure your safety. |
| H1 | Is the current scenario safe? | The current scenario is rated generally as high-risk due to foggy weather and the need to cross the road ahead without street lights. Foggy days can cause drivers of vehicles to have a lower visible distance and a greater potential threat when crossing the road. |
| H2 | Which landmarks in this scene would be most valuable for a blind pedestrian's navigation, why? | The traffic light at 3 meters straight ahead in the direction of your 11 o'clock serves as the most valuable landmark due to its size and position. The bus stop 8 meters far at your 2 o'clock offers secondary value with its shelter structure, providing tactile and spatial reference. The storefront signs on the left, while visible, have lower navigational value due to potential changes. |

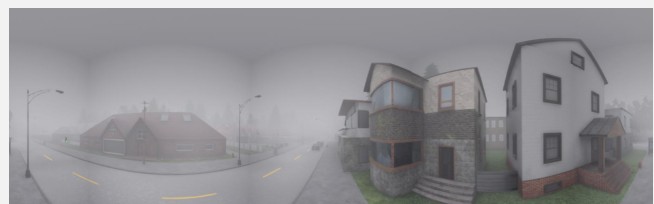

Source: *Mixed03_000182*
Q(E4): What is the color of the building at 3 o'clock?
A: Unknown.
**Ignorant**
**[revised] A: The building is white with dark roof tiles.**

Part of Drone View RGB Mixed03_000182

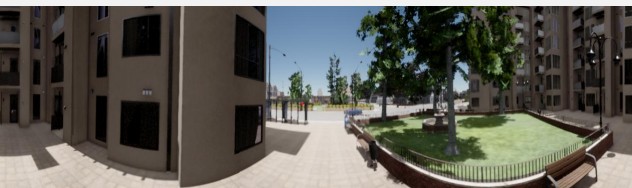

Source: *Plaza04_000160*
Q(M1): What is the relative position of the building to the bus stop?
A: Not Clear.
**Ignorant**
**[removed]**

Part of Walker View RGB Plaza04_000160

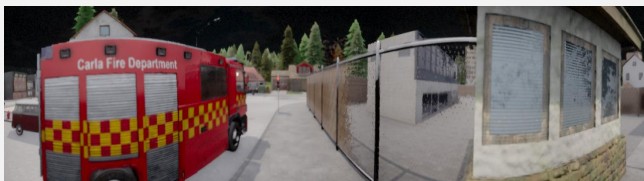

Source: *Parking04_000287*
Q(M3): Which view gives more information about vehicles?
A: The **rgb view** provides most.....
**Outrageous**
**[removed]**

Part of Walker View RGB Parking04_000287

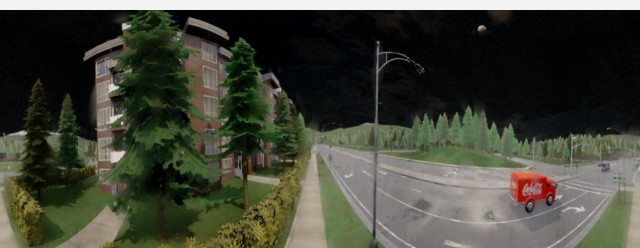

Source: *Corner18_000288*
Q(M3): Which view provides the best information on high obstacles?
A: The **depth** provide .....
**Outrageous**
**[removed]**

Source: *Corner18_000288*
Q(M3): What is the color of the truck on the road?
A: 0x8C0C08.
**Nonsense**
**[revised] A: Red**

Part of Drone View RGB Corner18_000288

Source: *Corner22_000180*
Q(M3): Which view gives more detailed information about the corner cases?
A: Question about corner cases.
**Nonsense**
**[removed]**

Source: *Gasstation07_000201*
Q(M3): Which view gives the most comprehensive layout of the gas station?
A: Views and Drone Views.
**Nonsense**
**[removed]**

Source: *Mixed02_000161*
Q(M3): Which view gives more information about dangers?
A: Question: Which view gives more......
**Nonsense**
**[removed]**

Figure 10: Examples of filtering work

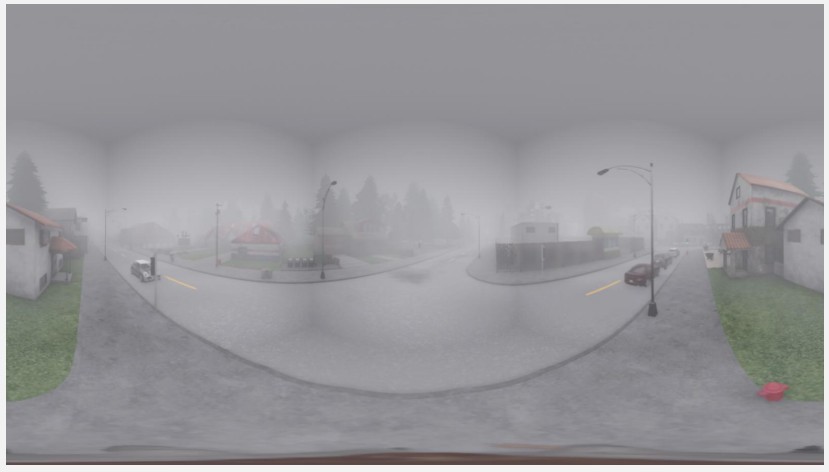

Source: *Mixed03_000716*
**Q(H1):** Is the current scenario safe?
**A:** The current scenario is rated as **high-risk** due to the **foggy weather**, combined with the **dangerous road crossing** ahead. The limited visibility increases the potential for unexpected hazards during the crossing.

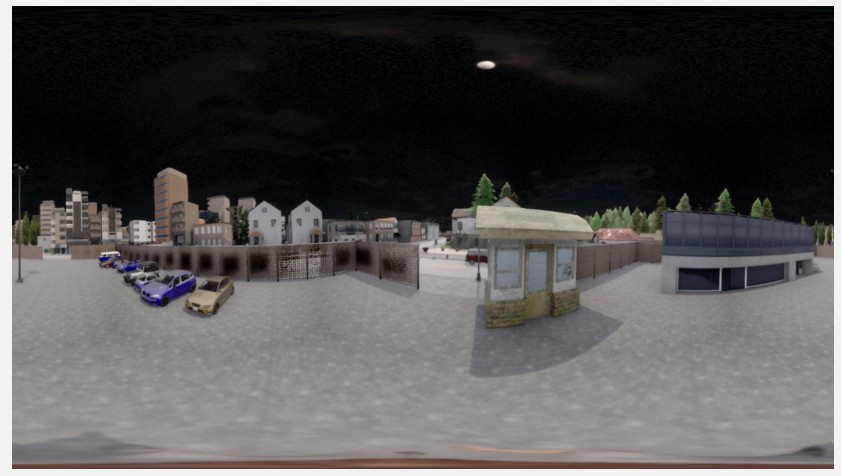

Source: *Parking04_000171*
**Q(H2):** Which landmarks in this scene would be most valuable for a blind pedestrian's navigation, and why?
**A:** The **gate at 2 o'clock** provides directional constancy and structural reliability crucial for navigation. With the curved kiosk nearby at 3 o'clock offers secondary value as a distinctive, immovable feature aiding orientation within the parking area.

Figure 11: Examples of hard level VQA pairs

Table 15. Resource List

| Resource | Content |
| --- | --- |
| CPUs | Intel Xeon Platinum 8368 |
| CPU Sockets per node | 2 |
| CPU Cores per node | 76 |
| CPU Threads per node | 152 |
| Cache L1 | 64K (per core) |
| Cache L2 | 1MB (per core) |
| Cache L3 | 57MB (shared, per CPU) |
| Main memory | 512 GB |
| Accelerators | 4x NVIDIA A100-40 |
| Memory per accelerator | 40 GB |
| Local disks | 960 GB NVMe SSD |
| Interconnect | InfiniBand HDR |

```
internvl/train/internvl_chat_finetune.py
  −−model_name_or_path "pretrained/InternVL2−8B"
  −−conv_style "internlm2−chat"
  −−force_image_size 448
  −−max_dynamic_patch 6
  −−down_sample_ratio 0.5
  −−freeze_llm True
  −−freeze_mlp True
  −−freeze_backbone True
  −−use_llm_lora 16
  −−vision_select_layer −1
  −−dataloader_num_workers 4
  −−bf16 True
  −−num_train_epochs 4
  −−per_device_train_batch_size
  −−gradient_accumulation_steps
  −−save_strategy "steps"
  −−save_steps 200
  −−save_total_limit 1
  −−learning_rate 4e−5
  −−weight_decay 0.05
  −−warmup_ratio 0.03
  −−lr_scheduler_type "cosine"
  −−logging_steps 1
  −−max_seq_length 4096
  −−do_train True
  −−grad_checkpoint True
  −−group_by_length True
  −−dynamic_image_size True
  −−use_thumbnail True
  −−ps_version 'v2'
  −−deepspeed "zero_stage1_config.json"
  −−report_to "tensorboard"
```

Figure 12: Finetune Parameter

```
messages = [
        {
                "role": "system",
                "content":
                        "You are an intelligent evaluator designed to evaluate the correctness and
                        similarity of generative outputs for question−answer pairs. "
                        "Your task is to compare the model prediction answer with the correct answer
                        and determine if they match in meaning. Here's the scoring criteria:\n\n"
                        "### Scoring Criteria:\n"
                        "5 = Perfect match or Correct in meaning\n"
                        "4 = Key information correct, minor flaws\n"
                        "3 = Partially correct\n"
                        "2 = Mostly wrong answer for key query, but some relevance\n"
                        "1 = Completely wrong or nonsense sentences\n\n"
                        "Your response must ONLY be the integer score (e.g., 4). DO NOT include
                        any text or explanation."
        },
        {
                "role": "user",
                "content":
                        f"Question: {question}\n"
                        f"Correct Answer: {gt_answer}\n"
                        f"Predicted Answer: {pred_answer}\n\n"
                        "Please provide a score from 1 to 5 based on how well the predicted answer
                        matches the correct answer."
        }
    ]
```

Figure 13: Evaluation Prompt

