# OpenReview forum: "mmWalk: Towards Multi-modal Multi-view Walking Assistance"
_NeurIPS.cc/2025/Datasets_and_Benchmarks_Track — NeurIPS 2025 Datasets and Benchmarks Track poster_

### Official Review · Reviewer_ZdR8 · 2025-06-19

**Rating:** 4
**Confidence:** 4

**Summary:**

This paper introduces mmWalk, a new simulated dataset for walking assistance for people with blindness or low vision. The dataset is multi-modal (RGB, depth, semantic) and multi-view (walker, drone, dog), featuring 120 trajectories with a focus on real-world corner cases. The authors also present mmWalkVQA, a benchmark with over 69k question-answer pairs to evaluate Vision-Language Models on navigational and safety tasks, showing that current models struggle but improve significantly after being fine-tuned on mmWalk.

**Additional Feedback:**

I suggest rethinking the evaluation for numerical answers by using direct parsing and comparison, as an LLM-as-judge can be unreliable for these tasks. To better prove the unique value of mmWalk, consider a control experiment where you compare a model finetuned on your dataset against one finetuned on another dataset like VizWiz. It would also strengthen the paper to explore and discuss more advanced fusion techniques for the multi-view and multi-modal data.

**Dataset Code Accessibility:**

Partly

**Dataset Code Comments:**

While the dataset and code are linked, I had some trouble with accessibility. The data files on the hosting platform were difficult to access, which could hinder reproducibility for the community.

**Ethical Comments:**

The use of a simulated environment is a strong point, as it proactively handles privacy and consent issues. To further improve the responsible release, the authors should add formal RAI metadata to provide more transparency.

**Ethical Considerations:**

No, there are no or only very minor ethics concerns

**Final Justification:**

The authors provided a comprehensive rebuttal that addressed most of my key concerns, particularly regarding the automated evaluation. Their new experiment showing a high correlation between human ratings and the GPT-4o judge significantly strengthens the validity of their benchmark. The paper introduces a novel and large-scale dataset for a crucial application area. However, significant limitations remain, namely that the dataset is purely synthetic and lacks any form of evaluation with the target user community (BLV individuals). While the work is a valuable first step, these points prevent a full-throated acceptance. The paper's value as a research resource outweighs these limitations, placing it firmly in the borderline accept category.

**Limitations Weaknesses:**

The paper has some issues with technical novelty, as it relies on integrating well-established tools like the CARLA simulator and GPT-4o rather than proposing new methods. Since the data is entirely synthetic, there's also a risk it doesn't capture the full diversity of real-world scenarios, which could limit how well the models generalize. Furthermore, its analysis is a bit confirmatory—demonstrating known VLM flaws and showing that fine-tuning improves performance are expected outcomes that offer limited new insight.

**Strengths Contributions:**

The paper tackles a very important and relevant problem: safe navigation for people with visual impairments. Its main strength lies in the comprehensive dataset, which combines multi-view, multi-modal, and temporal data while focusing on critical corner cases like uneven paths and finding entrances. Additionally, the structured VQA benchmark is a significant contribution, allowing for a fine-grained analysis of model capabilities beyond generic question answering.

---

> ### Author Rebuttal · Authors · 2025-07-30
>
> Dear Reviewer ZdR8,
>
> Thank you very much for your thoughtful and valuable comments. We are particularly grateful that you recognized the importance of safe navigation for people with visual impairments, as well as the strengths of our multi-modal, multi-view dataset and fine-grained VQA benchmark.
>
> We will address your concerns regarding technical novelty, real-world applicability, and benchmark design. While we understand your reservations, we sincerely hope the following clarifications demonstrate the novelty and relevance of our contributions.
>
> **[W1]Paper has issues with technical novelty**
> > The paper has some issues with technical novelty, as it relies on integrating well-established tools like the CARLA simulator and GPT-4o rather than proposing new methods.  Furthermore, its analysis is a bit confirmatory—demonstrating known VLM flaws and showing that fine-tuning improves performance are expected outcomes that offer limited new insight.
>
> At first step, we respectfully disagree with the notion that our work lacks novelty. While our implementation leverages well-established tools like the CARLA simulator and GPT-4o, our contribution lies not in the tools themselves, but in how we design and structure a large-scale, multi-view, multi-modal dataset tailored for real-world BLV outdoor navigation challenges.
>
> To the best of our knowledge, mmWalk is the first dataset to:
> * Combine three egocentric perspectives (walker, guide dog, and drone) with synchronized RGB, depth, and semantic segmentation;
>
> * Incorporate BLV-specific corner cases and accessibility landmarks, including blind intersections, unmarked road edges, and terrain transitions, which are not represented in existing datasets;
>
> * Provide a structured benchmark (mmWalkVQA) designed to assess spatial reasoning, perception-level understanding, and risk awareness across multiple difficulty levels.
>
>
> **[W2] Entirely Synthetic Dataset**
> >Since the data is entirely synthetic, there's also a risk it doesn't capture the full diversity of real-world scenarios, which could limit how well the models generalize.
>
> We agree that synthetic datasets have inherent limitations in ecological realism. However, mmWalk was designed as a scalable first step toward capturing the diversity and spatial complexity of outdoor environments relevant to BLV navigation, integrating BLV corner cases and landmarks. We believe mmWalk offers a unique and valuable contribution to the BLV community. Furthermore, to begin bridging the sim-to-real gap, we finetuned InternVL2-8B on mmWalk and evaluated it on EgoTextVQA, demonstrating early signs of transferability. We explicitly discuss the limitations of simulation in the paper, and extending mmWalk with real-world data is a key direction we are actively pursuing.
>
> **[W3] Experiment Design**
> >Furthermore, its analysis is a bit confirmatory—demonstrating known VLM flaws and showing that fine-tuning improves performance are expected outcomes that offer limited new insight.
>
>
> To this concern, we respectfully clarify that our work primarily contributes a new dataset and benchmark tailored to underrepresented, safety-critical scenarios in the BLV context, rather than proposing new model architectures or conducting a comprehensive study of VLMs. While we include finetuning results to demonstrate the utility and challenge of mmWalkVQA, a full-scale comparison across more models is beyond the scope of this paper. That said, we believe mmWalkVQA provides a strong foundation for future research to systematically evaluate and improve VLMs in outdoor BLV-related tasks.
>
> Beyond standard performance reporting, we respectfully argue that our analysis also includes fine-grained evaluations enabled by the structure of mmWalk and additional experiments. These include comparisons across different scenarios, ablations over different views (e.g., walker, drone, guide dog), and a sim-to-real experiment, all of which highlight the unique insights that mmWalk and mmWalkVQA can offer. We found that existing MLLMs lack the capability to leverage complementary views, whereas training with our dataset enhances their ability to take advantage of these compensatory perspectives.
>
> **[Additional Feedback]**
> > I suggest rethinking the evaluation for numerical answers by using direct parsing and comparison, as an LLM-as-judge can be unreliable for these tasks. To better prove the unique value of mmWalk, consider a control experiment where you compare a model finetuned on your dataset against one finetuned on another dataset like VizWiz. It would also strengthen the paper to explore and discuss more advanced fusion techniques for the multi-view and multi-modal data.
>
> Thank you for your thoughtful comments! We fully understand your concerns around LLM-based evaluation. However, LLM-as-a-judge has become a widely adopted practice in recent works (e.g., [EgotextVQA], [GuideDog]), offering scalability and consistency across large QA sets.
>
> To address reliability against LLM as a judge, we have recently conducted an additional human rating phase. Specifically, we randomly sampled 10% of VQA pairs from five model outputs covering a wide range of model types: Finetuned InternVL2-8B, Qwen2 in zero-shot and 3-shot settings, LLaVA-OneVision in 3-shot setting，LLaVA-Next in zero-shot. In total, 3,575 VQA samples across all QA categories were rated by human annotator. We then calculated Spearman’s ρ between the human scores and GPT-4o automatic scores to evaluate correlation. Our results show:
> * The finetuned model (InternVL2-8B) achieved a Spearman’s ρ = 0.924, indicating a very high agreement between human and GPT scores.
>
>
> * Across all five models, the average Spearman’s ρ was 0.864, demonstrating the overall reliability and consistency of our automatic evaluation pipeline.
>
>
> We also appreciate the suggestion you have proposed. In fact, we have conducted a cross-dataset evaluation using VizWiz, where we compared a model fine-tuned on mmWalk with one fine-tuned on VizWiz itself. While these results were not included in the main paper we present the omitted segment and performance table here.
>
>
> | Model                         | Inference On            | Input | Overall Accuracy | BLEU-4 | METEOR |
> | ----------------------------- | ----------------------- | ----- | ---------------- | ------ | ------ |
> | mmWalk finetuned              | VizWiz Validation Split | Question + Image      | 0.325            | 0.054  | 0.288  |
> | VizWiz unfinetuned            | VizWiz Test Split       | Question + Image      | 0.137       | 0       |0.078
> | VizWiz finetuned              | VizWiz Test Split             |Question + Image     | 0.466            | 0.314  | 0.297  |
>
>
> To clarify, we note that VizWiz is a valuable dataset, but its focus is primarily on general BLV assisting questions, mostly indoor, whereas mmWalk is designed around outdoor walking scenes, navigation-related challenges and unique corner cases for BLV community. As such, we did not refer this additional cross dataset evaluation experiment in our submitted paper.
>
> Moreover, as you mentioned, advanced fusion strategies are indeed important for unlocking the full potential of multi-view and multi-modal data. mmWalk’s current benchmark uses off-the-shelf VLMs, but we envision future iterations of the dataset incorporating dedicated fusion baselines and serving as a testbed for architectural innovations in multi-view perception.
>
> **[Dataverse accessibility]**
> > While the dataset and code are linked, I had some trouble with accessibility. The data files on the hosting platform were difficult to access, which could hinder reproducibility for the community.
>
> We are sorry for any difficulty you experienced while accessing the dataset. We currently follow the official NeurIPS submission policy, using Harvard Dataverse for dataset submission. All dataset metadata and instructions are included in the repository. That said, we will be working to mirror the dataset on additional public platforms with simplified download options for broader accessibility.
>
> We hope these clarifications help address your concerns. We remain confident that mmWalk makes a meaningful contribution by providing a structured, safety-relevant, multi-view benchmark for assessing and improving vision-language models in accessibility-related navigation.
>
> Please let us know if you have any further concerns and questions.
>
> Sincerely,
>
> Authors
>
> **References**
> [EgoTextVQA]: Sheng Zhou, Junbin Xiao, Qingyun Li, Yicong Li, Xun Yang, Dan Guo, Meng Wang, Tat-Seng Chua, & Angela Yao. (2025). EgoTextVQA: Towards Egocentric Scene-Text Aware Video Question Answering.
>
> [GuideDog]: Junhyeok Kim, Jaewoo Park, Junhee Park, Sangeyl Lee, Jiwan Chung, Jisung Kim, Ji Hoon Joung, & Youngjae Yu. (2025). GuideDog: A Real-World Egocentric Multimodal Dataset for Blind and Low-Vision Accessibility-Aware Guidance.

---

> > ### Comment · Reviewer_ZdR8 · 2025-08-02
> >
> > I thank the authors for their thorough and constructive rebuttal. I especially appreciate the new human evaluation experiment, which has successfully addressed my primary concern regarding the potential bias of the automated GPT-4o evaluation. The clarifications on the experimental setup and dataset structure have also improved the paper's clarity.
> >
> > While the dataset remains purely synthetic, the paper stands as a valuable and well-designed contribution to an important research area. I am now more confident in the work's contributions and support its acceptance. I strongly encourage the authors to pursue real-world validation and user studies in their future work to bridge the sim-to-real gap.

---

> > > ### Author Response · Authors · 2025-08-02
> > >
> > > Thank you very much for your feedbacks. We are glad that our response addressed your concerns.
> > >
> > > We fully agree with your follow-up suggestions. More real-world validation and user studies will not only bridge the gap between simulation and reality, but also enrich the content of our work, strengthen our subsequent contributions, and enhance the practicality and comprehensiveness of mmwalk. This is a very valuable and exciting direction for the future work.
> > >
> > > Thank you again for your thoughtful feedback and consideration.
> > >
> > > Sincerely,
> > >
> > > Authors

---

### Official Review · Reviewer_PaiM · 2025-06-30

**Rating:** 4
**Confidence:** 4

**Summary:**

This paper introduces mmWalk, a novel simulated dataset designed to support multi-modal, multi-view visual assistance for individuals with blindness or low vision (BLV) in outdoor environments. The dataset comprises over 559k panoramic images with RGB, depth, and semantic segmentation modalities, collected from three viewpoints—walker, guide dog, and drone. Additionally, the authors present mmWalkVQA, a large-scale visual question answering (VQA) benchmark with over 69k question-answer pairs across nine task types, encompassing both perception-level and reasoning-level challenges (e.g., risk assessment, landmark identification).
The paper further evaluates several state-of-the-art vision-language models (VLMs) under zero-shot and few-shot settings, demonstrating their limitations in safety-critical and navigation tasks. A fine-tuned version of InternVL2-8B on mmWalkVQA shows significant performance gains, both on the proposed benchmark and on real-world VQA datasets such as EgoTextVQA.

**Dataset Code Accessibility:**

Yes

**Ethical Considerations:**

No, there are no or only very minor ethics concerns

**Final Justification:**

The paper presents a well-motivated dataset with thoughtful design. While my concerns have been addressed, some areas—such as multi-view effectiveness and evaluation transparency—could benefit from further refinement in the final version.

**Limitations Weaknesses:**

1. Synthetic-Only Dataset Limits Real-World Generalization:

Although the dataset is meticulously constructed within a simulated environment, it lacks real-world data. As such, it may not capture the full variability, unpredictability, and sensor noise present in outdoor environments. This poses a limitation on the ecological validity and generalizability of the proposed benchmark, despite some promising results from cross-dataset evaluations.

2. Absence of User-Centered Evaluation or Human-Subject Studies:

The system has not been evaluated with actual users from the BLV (Blind and Low Vision) community or accessibility practitioners. Without qualitative or quantitative feedback from target users, the practical relevance and usability of the generated answers remain speculative. A human-in-the-loop evaluation would significantly strengthen the claims.

3. Static Image-Based VQA Falls Short for Dynamic Navigation Contexts:

Real-world navigation is inherently dynamic and sequential, often involving temporal reasoning and real-time decision-making. The reliance on static image-based VQA limits the applicability of the system in realistic scenarios where motion cues, action planning, or trajectory updates are essential.

4. Marginal Gains from Multi-View Fusion Suggest Redundancy:

In Table 7, the performance gains from incorporating additional viewpoints (e.g., dog and drone views) over the walker view are relatively small. This raises concerns about information redundancy or suboptimal fusion strategies. A deeper ablation or modality contribution analysis would help clarify this issue.

5. Evaluation Bias Due to Fully Automated GPT-4o Scoring:

The generation and evaluation of all VQA pairs are fully automated using GPT-4o, which introduces potential biases due to model self-alignment and lack of independent ground truth. The absence of human-labeled gold answers or third-party evaluators undermines the objectivity and reliability of the reported results. Incorporating human ratings or inter-rater agreement analysis would increase the robustness of the benchmark.

**Strengths Contributions:**

1. Comprehensive and Well-Motivated Dataset Design:

The dataset is thoughtfully designed around practical challenges encountered by BLV individuals. It includes 8 types of BLV-relevant corner cases (e.g., uneven terrain, road crossing without signals) and 18 accessibility landmarks drawn from ATmaps standards. This aligns well with real-world needs.

2. Novel Multi-Modal, Multi-View Structure:

mmWalk uniquely combines egocentric RGB, depth, and semantic views from three perspectives (walker, dog, drone), enabling rich scene understanding from diverse vantage points. The panoramic projection and synchronized metadata further enhance its utility for spatial reasoning tasks.

3. Thorough Benchmarking and Task Definition:

The mmWalkVQA benchmark includes a diverse set of question types with well-structured difficulty levels (easy, medium, hard). The use of LLM-based evaluation and comparison across multiple SOTA models offers valuable diagnostic insight into current limitations of VLMs for assistive navigation.

---

> ### Author Rebuttal · Authors · 2025-07-30
>
> Dear Reviewer PaiM,
>
> Thank you very much for your thoughtful and valuable comments. We’re especially grateful that you found our dataset design, multi-view structure, and benchmarking efforts to be comprehensive. We also thank you for raising important concerns regarding real-world generalization, user-centered evaluation, and dataset limitations. Please find our detailed responses below.
>
> **[W1] Synthetic-Only Dataset**
> > Although the dataset is meticulously constructed within a simulated environment, it lacks real-world data. As such, it may not capture the full variability, unpredictability, and sensor noise present in outdoor environments. This poses a limitation on the ecological validity and generalizability of the proposed benchmark, despite some promising results from cross-dataset evaluations.
>
> We recognize that real-world environments exhibit greater variability, unpredictability, and sensor noise, which are indeed difficult to replicate in simulation. We sincerely thank the reviewer for raising this important point, which helpfully identifies a meaningful direction for future extensions of our work.
>
> To clarify, mmWalk represents our first step toward building large-scale multi-modal datasets tailored for BLV navigation. While we acknowledge that deploying such a multi-view, multi-agent, multi-modal setting in real-world environments remains technically and economically challenging for us at present, we argue that mmWalk provides a unique and valuable contribution by offering over 559k panoramic frames across RGB, depth, and semantic modalities, along with richly annotated BLV-specific corner cases and landmarks. mmWalk paves the way for the development and evaluation of next-generation multi-agent assistive systems.
>
> Importantly, we reiterate our agreement that incorporating real world multi-view data would further enhance ecological validity. Expanding mmWalk into a real-world version is indeed one of our long-term goals.
>
>
> **[W2] Absence of User-Centred**
> > The system has not been evaluated with actual users from the BLV (Blind and Low Vision) community or accessibility practitioners. Without qualitative or quantitative feedback from target users, the practical relevance and usability of the generated answers remain speculative. A human-in-the-loop evaluation would significantly strengthen the claims.
>
> We would like to clarify that the current mmWalk is not positioned as a deployable system, but rather as a research dataset and benchmark aimed at facilitating the development and evaluation of VLMs in BLV relevant outdoor navigation tasks. Consequently, we do not yet have the infrastructure or access to conduct large-scale feedback collection from target BLV users.
>
> Notably, the motivation and early design of mmWalk dataset are user-centric, as we conducted background research that included interviews and consultations with experts in accessibility as co-authors and two additional experts who we will acknowledge, who provided critical guidance, especially on many of the corner cases represented in our dataset, which helped us ground the dataset in real-world user challenges even within a simulated framework.
>
> We agree that future work incorporating direct human-in-the-loop evaluation, particularly with BLV users, would further strengthen the practical impact of mmWalk, and we are actively exploring ways to collaborate with accessibility communities in the next phase of this research.
>
> **[W3] Static VQA Shortcoming**
> > Real-world navigation is inherently dynamic and sequential, often involving temporal reasoning and real-time decision-making. The reliance on static image-based VQA limits the applicability of the system in realistic scenarios where motion cues, action planning, or trajectory updates are essential.
>
> We agreed that static VQA has some shortcomings. However, we would like to clarify that mmWalk is inherently not fully static dataset. Each trajectory in mmWalk consists of an ordered sequence of frames, sampled at consistently, and reflects a continuous walking path rather than isolated or randomly sampled images. In addition, many of the trajectories include dynamic agents, such as moving vehicles or pedestrians, which are captured across multiple frames from different viewpoints. While our current benchmark is structured around frame-level VQA tasks, we acknowledge the importance of richer temporal modeling. To this end, we will provide stitched video visualizations of each trajectory as metadata, which will provide additional context and support a more reliable navigation scenario.
>
> **[W4] Concern multi-view redundancy**
> > In Table 7, the performance gains from incorporating additional viewpoints (e.g., dog and drone views) over the walker view are relatively small. This raises concerns about information redundancy or suboptimal fusion strategies. A deeper ablation or modality contribution analysis would help clarify this issue.
>
>
> | Input       | Overall Score | Gains |
> | ----------- | ------------- | ----- |
> | Walker Only          | 53.19        |       |
> | Walker + Dog         | 53.85        |  +0.66     |
> | Walker + Drone       | 54.08        |  +0.89     |
> | Walker + Dog + Drone | 55.05        |  +1.86     |
>
> To address your concern, we highlight here a variant of Table 7, focusing on finetuned InternVL2-8B model, where we isolate the walker view as a baseline. This enables a clearer observation of the marginal benefit of adding single views in input, which supports our claim that the multi-view structure in mmWalk is functionally beneficial, and that models can indeed learn to leverage spatially complementary information across viewpoints when trained appropriately.
>
> Regarding your concern that multi-view input may introduce redundancy or reflect suboptimal input strategy, we respectfully disagree. Our ablation shows that finetuned models are capable of utilizing the multi-view inputs. In contrast, zero-shot models exhibit larger inconsistency, with some categories showing higher scores for single-view inputs (as shown in original Table 7). We attribute this to the limited capacity of existing VLMs to perform spatial information integration in a zero-shot setting, rather than to redundancy in the data itself.
>
> Yet, we agree that conducting a fine-grained contribution analysis of each view over each VQA category would provide further insights into model behavior. This is a valuable direction we would like to explore in future work, especially as stronger multi-modal fusion models become available.
>
>
> **[W5] Evaluation Bias**
> > The generation and evaluation of all VQA pairs are fully automated using GPT-4o, which introduces potential biases due to model self-alignment and lack of independent ground truth. The absence of human-labeled gold answers or third-party evaluators undermines the objectivity and reliability of the reported results. Incorporating human ratings or inter-rater agreement analysis would increase the robustness of the benchmark.
>
> Based on your valuable suggestion, we have recently conducted an additional human rating phase, inspired by the methodology in [Open-EQA]. Specifically, we randomly sampled 10% of VQA pairs from five model outputs covering  model types: Finetuned InternVL2-8B, Qwen2 in zero-shot and 3-shot settings, LLaVA-OneVision in 3-shot setting and LLaVA-Next in zero-shot.
>
> In total, 3,575 VQA samples across all QA categories were rated by human annotator. We then calculated Spearman’s ρ between the human scores and GPT-4o automatic scores to evaluate correlation. Our results show:
>
> * Across all five models, the average **Spearman’s ρ was 0.864**, demonstrating the overall reliability and consistency of our automatic evaluation pipeline.
>
> * Our finetuned model (InternVL2-8B) achieved a spearman’s ρ = 0.924, indicating a very high agreement between human and GPT scores.
>
> We will include these supplementary analyses and results in the final version of the paper and the appendix to further enhance transparency and support the reproducibility of our work. Moreover, incorporating human-annotated gold answers and conduct extensional human-based evaluation is indeed a meaningful direction for future work.
>
> Please let us know if you have any further concerns and questions.
>
> Sincerely,
>
> Authors
>
> **Reference**
>
> [OpenEQA] A. Majumdar et al., "OpenEQA: Embodied Question Answering in the Era of Foundation Models," 2024 IEEE/CVF Conference on Computer Vision and Pattern Recognition (CVPR), Seattle, WA, USA, 2024, pp. 16488-16498

---

> > ### Comment · Reviewer_PaiM · 2025-08-05
> >
> > Thank you for your response. All of my concerns have been addressed. However, I hope the issues of limited multi-view gains and missing human evaluation details (e.g., number of raters, evaluation guidelines) can be further clarified in the reviewed paper or future work. I will maintain my current score.

---

> > > ### Author Response · Authors · 2025-08-05
> > >
> > > Thank you very much for your feedbacks. We are glad that our response addressed your concerns.
> > >
> > > We will make sure to clarify the human evaluation details in the paper, including the samples, number of raters and full evaluation guidelines. We also plan to further explore the gains from additional views and conduct more related experiments in our future work.
> > >
> > > Sincerely,
> > >
> > > Authors

---

### Official Review · Reviewer_u3mH · 2025-07-01

**Rating:** 5
**Confidence:** 4

**Summary:**

This paper presents mmWalk, a novel simulated dataset for multi-modal, multi-view walking assistance focused on the Blind and Low Vision (BLV) community. It provides panoramic RGB, depth, semantic images from walker, dog, and drone perspectives across 120 trajectories, annotated with BLV-relevant corner cases and navigational landmarks. A corresponding benchmark, mmWalkVQA, includes 69k VQA pairs generated with GPT-4o and evaluated using GPT-4o-mini as judge. The dataset aims to promote safer, more informed autonomous navigation and assistive AI.

**Dataset Code Accessibility:**

Yes

**Ethical Comments:**

No.

The dataset is entirely simulated and does not involve human subjects, privacy-sensitive data.

**Ethical Considerations:**

No, there are no or only very minor ethics concerns

**Limitations Weaknesses:**

1. Missing or insufficient experiments:

(1) Lack of GPT-4o baseline: The comparison of different VLMs omits GPT-4o, which is used to generate ground truth answers. This raises the question of whether the task is inherently already solved by GPT-4o. If that is the case, the need for a new dataset becomes less compelling.

(2) Limited finetuning analysis: While the paper demonstrates that finetuning InternVL2-8B on mmWalk improves performance, this single comparison is somewhat limited. Including additional VLMs in the finetuning experiments would strengthen the argument (though this is not strictly necessary for acceptance).

2. Lack of clarity in descriptions:

(1) Ambiguity of the term “frame”: The term “frame” is frequently used in the dataset creation section, but it is unclear what it specifically refers to. How does a frame relate to a single image or a panoramic image? Clarifying this would improve the readability.

(2) VQA filtering details: The paper mentions filtering out low-quality QA pairs, and manually correcting some of them. However, it lacks statistics on how many pairs were manually modified. This information would help quantify both the human effort involved and the limitations of GPT-4o in generating high-quality data.

(3) Lack of input specification for Tables 5 and 6: The experimental setup for Tables 5 and 6 does not clearly state what the model inputs were (e.g., which views or modalities were used). This lack of detail makes it difficult to fully interpret the results and reproduce the experiments.

**Strengths Contributions:**

1.	This paper focuses on the task of BLV navigation, which has both scientific and societal value.
2.	The overall quality of the paper is high. This is reflected in several aspects: the paper is well-structured; the introduction section is thorough and clearly emphasizes the importance of the task; the dataset collection process is comprehensively described; the dataset itself is publicly available on Harvard Dataverse, enriched with detailed annotations and a large set of VQA triplets. In addition, the dataset provides diverse modalities, which support a wide range of downstream research.
3.	The experimental design is relatively thorough. It evaluates a variety of state-of-the-art VLMs, demonstrating that their VQA capabilities vary in terms of scene understanding. It also shows that training on this dataset can lead to noticeable performance improvements.

---

> ### Author Rebuttal · Authors · 2025-07-30
>
> Dear Reviewer u3mH,
>
> Thank you very much for your thoughtful and valuable feedback. We’re especially grateful that you found our work to be of high overall quality with good structure and richness in experiments. We also appreciate your comments regarding our experimental design and pointing out unclear terms in paper. We will address each of your concerns in detail below.
>
> **[W1.1] Lack of GPT-4o baseline**
> > Lack of GPT-4o baseline: The comparison of different VLMs omits GPT-4o, which is used to generate ground truth answers. This raises the question of whether the task is inherently already solved by GPT-4o. If that is the case, the need for a new dataset becomes less compelling.
>
> We intentionally exclude GPT-4o from the benchmark for the following reasons. Since GPT-4o is involved in both the question-answer generation and the evaluation process, including it as a test-time baseline would lead to circular evaluation and introduce a systematic bias. Additionally, our goal is to compare VLMs that were not involved in data construction, ensuring a fair and unbiased assessment of general VLM capabilities on mmWalkVQA. Respectfully, we believe this design decision strengthens, rather than weakens, our experiment results.
>
> As to the question of whether GPT-4o has already “solved” the task, we respectfully argue that, despite we used GPT-4o as QA generator, which serves as ground truth in our benchmark, it does not actually “solve” the task. The performance of generated VQA, although acceptable, remains far from "solved", as shown in Sec. 3.2 Table 3, term 'Answer Correctness' has received 4.62/5 in human rating sampling phase, while term 'Answer Actionability'，i.e. whether it is assumed that users can safely take the next step according to the instructions in answer, was only scored 4.43/5.
>
> **[W1.2] Limited finetuning analysis**
> >Limited finetuning analysis: While the paper demonstrates that finetuning InternVL2-8B on mmWalk improves performance, this single comparison is somewhat limited. Including additional VLMs in the finetuning experiments would strengthen the argument (though this is not strictly necessary for acceptance).
>
>
> We agree that additional fine-tuning of the model is necessary if more comparative tests are to be conducted for VLM. However, we would like to clarify that the finetuning of InternVL2-8B in our work is intended to support a specific sim-to-real transferability experiment, rather than to perform a comprehensive comparison of finetuned VLMs. The reason for choosing InternVL2-8B was that it performed best on our primary experiment, as presented in Sec. 4.3.
>
> The main focus in this work is on the construction and analysis of the mmWalk dataset and benchmark, which is designed to evaluate the reasoning capabilities of state-of-the-art VLMs across multi-view outdoor navigation scenarios. The inclusion of InternVL2-8B finetuning serves as a case study to demonstrate how the mmWalkVQA benchmark can be used to improve downstream sim-to-real performance on real-world BLV oriented tasks. Therefore, our goal was not to conduct exhaustive comparisons between different finetuned models, but rather to illustrate the utility of mmWalkVQA as a training resource.
>
> We reiterate our agreement that exploring additional model level innovations and broader finetuning comparisons (e.g., across multiple VLM families) would be a valuable extension of this work. We plan to include these directions in future versions of the mmWalk benchmark suite.
>
> **[W2.1] Ambiguity of term “frame”**
> > Ambiguity of the term “frame”: The term “frame” is frequently used in the dataset creation section, but it is unclear what it specifically refers to. How does a frame relate to a single image or a panoramic image? Clarifying this would improve the readability.
>
>
> Thank you very much for pointing this out. To begin, we would like to answer your question that all images in our mmWalk dataset are panoramas. In our manuscript, the term frame refers to a single timestamp, at which multiple panoramic images (from different views) are captured, including all modalities. More specifically, a frame consists of synchronized panoramic rgb, depth and semantic segment images captured from walker, drone, and dog views. In contrast, 'image' is used to refer to an individual panorama from one specific view. We acknowledge that this distinction was not clearly articulated in the current version of the manuscript. Furthermore, the usage of term “frame” in Section 4.5 is adopted directly from the EgoTextVQA paper, where a frame is also defined as the image in one timestamp sampled from a video.
>
> We will revise the paper to make this terminology clearer, and explicitly define the relationship between frame and image in the context of our dataset.
>
> **[W2.2] VQA Filtering Details**
> > VQA filtering details: The paper mentions filtering out low-quality QA pairs, and manually correcting some of them. However, it lacks statistics on how many pairs were manually modified. This information would help quantify both the human effort involved and the limitations of GPT-4o in generating high-quality data.
>
> We fully agree that reporting the proportion of human intervention in automatically generated QA pairs is important for transparency, especially in VQA datasets relying on LLM-generated content.
>
> As described in Section 3.2, we implemented a keyword filtering phase to eliminate meaningless QA pairs. In this step, a total of 102 QA pairs were removed. In our second round human filtering, over 100 frames were sampled and approximate 70 qa pairs of sampled frames are manually corrected or edited, mostly in hard level, to improve clarity or correctness. We regret that we failed to recall and report the concret number in this phase, though some human fix examples can be found in Appendix. However, to further ensure the quality of mmWalkVQA, we performed a third round human validation, where a subset of the QA pairs was manually scored based on relevance and answer correctness. These results are presented in Table 3 of Section 3.2, showing that the final dataset maintains high quality overall.
>
> We believe these combined steps including automated filtering, human correction, and validation help uphold the integrity of mmWalkVQA. While we acknowledge the omission of specific statistics, we emphasize that the dataset quality is robust and does not compromise the strength of our contributions.
>
> **[W2.3] Lack of input specification**
> > Lack of input specification for Tables 5 and 6: The experimental setup for Tables 5 and 6 does not clearly state what the model inputs were (e.g., which views or modalities were used). This lack of detail makes it difficult to fully interpret the results and reproduce the experiments.
>
> We appreciate the reviewer’s suggestion regarding the explicit specification of input modalities in Table 5 and Table 6. Presenting the input modalities alongside the experimental results is indeed meaningful for better clarity and reproducibility. We will incorporate this improvement in future versions of the paper.
>
> To clarify: the baseline experiments in both Table 5 and Table 6 are conducted using the full multi-view input, which includes RGB images from all three perspectives.
>
> Please let us know if you have any further concerns and questions.
>
> Sincerely,
>
> Authors

---

> > ### Comment · Reviewer_u3mH · 2025-08-06
> >
> > Thank you to the authors for the detailed and constructive responses. I accept the authors’ explanations regarding the concerns on "limited finetuning" and the "absence of a GPT-4o baseline", and I will maintain my original rating recommending acceptance. I encourage the authors to improve the clarity of their descriptions.  I believe this dataset is valuable and, like other reviewers, I look forward to future extensions to real-world scenes.

---

> > > ### Author Response · Authors · 2025-08-06
> > >
> > > Thank you very much for your feedback. We are glad that our response addressed your concerns.
> > >
> > > We appreciate your high evaluation of mmWalk and we will improve the description clarity in the final version, we are also very excited about extending mmWalk to hybrid version with real-world scene in future work.
> > >
> > > Sincerely,
> > >
> > > Authors

---

> ### Comment · Area_Chair_R6Ez · 2025-08-08
> **Final Justification**
>
> Dear Reviewer u3mH,
>
> It looks like the “Mandatory Acknowledgement” has been submitted, but the “Final Justification” has not yet been completed.  Please fill the "Final Justification" by the deadline.
>
> Best,
> AC

---

### Official Review · Reviewer_Xgwu · 2025-07-03

**Rating:** 5
**Confidence:** 4

**Summary:**

This work introduces a multimodal multi-view walking assistance dataset called **mmWalk**, which is designed to help blind and low vision (BLV) users navigate safely in complex or extreme environments. The dataset integrates multi-view sensors and obstacle-free features, contains 120 trajectory paths among 7 scenario categories and 5 weather conditions, summing up 2.5M single images in total, covering multiple modalities such as RGB, depth, and semantics.

For the evaluation of Visual Question-Answering (VQA), this work further builds **mmWalkVQA**, which uses GPT-4o to generate and filter VQA triplets, resulting in a final set of 69,391 VQA triplets under 3 difficulty levels and 9 VQA types.

The evaluation of existing VLMs (LLaVA-OV, LLaVA-Next, Qwen2VL, InternVL2, Janus-Pro, Chameleon) on mmWalkVQA highlights notable shortcomings in their capacity to reason about spatial relationships, detect hazards, and understand multi-view scenes from the perspective of BLV users. By fine-tuning InternVL2 on the mmWalkVQA dataset, the overall performance is significantly enhanced.

**Dataset Code Accessibility:**

Yes

**Dataset Code Comments:**

The code and the hosted data are both accessible and well-organized.

**Ethical Comments:**

This work employs a simulation environment for data collection, addressing data security and privacy concerns. Additionally, the authors provide a detailed discussion regarding the limitations and potential societal impacts of their work.

**Ethical Considerations:**

No, there are no or only very minor ethics concerns

**Final Justification:**

The authors' rebuttal has addressed most of my concerns, so I am maintaining my original score and recommending acceptance. The dataset introduced in this work encompasses multi-view and multi-modal motion perception data, which will make a positive impact on research areas such as embodied intelligence and spatial understanding. However, the dataset is limited to outdoor scenes, and as pointed out by other reviewers, it lacks sim-to-real generalization and human subject studies, which may hinder its practical application in real-world scenarios.

**Limitations Weaknesses:**

1. The mmWalk dataset is limited to outdoor scenes, which may restrict its applicability to indoor environments and other varied settings.

2. In Table 8, some results are marked as *n.a.*, but there is no explanation for these missing values.

**Strengths Contributions:**

1. This work is driven by a clear and meaningful motivation. The outdoor navigation challenge is crucial, not only for assisting visually impaired and low-vision (BLV) individuals in their mobility but also for advancing spatial intelligence research.

2. The proposed mmWalk dataset is comprehensive and large-scale, encompassing a variety of perspectives, modalities, scenes, and weather conditions, with a total of over 559,000 panoramic images. This extensive dataset is poised to significantly advance research in visual navigation and spatial intelligence.

3. The constructed mmWalkVQA dataset is used to evaluate a series of existing state-of-the-art VLMs, providing an extensive and systematic benchmark.

4. The paper is well-written and organized clearly.

---

> ### Author Rebuttal · Authors · 2025-07-30
>
> Dear Reviewer Xgwu,
> Thank you very much for your thoughtful and valuable feedback. We're very grateful you found our work well-motivated, recognizing our comprehensive mmWalk dataset, mmWalkVQA benchmark for outdoor navigation and spatial intelligence, and clear writing. We will address your concern regarding indoor generalization and clarify the missing value in Table 8 in detailed responses below.
>
> **[W1] Limitation to indoor scenes**
> > The mmWalk dataset is limited to outdoor scenes, which may restrict its applicability to indoor environments and other varied settings.
>
> We fully agree that indoor and outdoor scenes differ substantially, the generalization across these domains is an important research direction.
>
> Indoor scene understanding and BLV assistance systems often rely on entirely different modalities (e.g., floor-plan priors, fine-grained object detection, or WiFi localization) [NaVIP, InCrowd-VI, Turn-by-turn Indoor] and typically benefit less from multi-agent fusion due to spatial constraints and limited visual variability. In contrast, our goal in mmWalk is to focus specifically on the outdoor domain, which presents a distinct set of real-world challenges. Thus, compared with indoor works, our mmWalk dataset includes longer trajectories, different weather conditions, complex BLV corner cases and landmarks. Additionally, as we introduced in Sec. 2 and highlighted in experiments in Sec. 4.4, multi-view sensing plays an important role in outdoor contexts where occlusion, uneven ground, and obstacles of varying height are the commonplace.
>
> That said, we deeply appreciate the reviewer's suggestion, and we acknowledge that extending our approach to indoor domains is a highly meaningful and exciting direction in the long term and will discuss it in the limitation and future work.
>
> **[W2] Missing Explanation in Table 8**
> > In Table 8, some results are marked as n.a., but there is no explanation for these missing values.
>
> Thank you so much! We will update our manuscript to clarify this in the caption of Table 8. Specifically, while EgoTextVQA provides detailed results per QA categoriy when using video inputs, it does not release corresponding scores for frame input. Thus, only the overall performance was reported in this case.
>
>
> Please let us know if you have any further concerns and questions.
>
> Sincerely,
>
> Authors
>
> **Reference:**
>
> [NaVIP]:Jun Yu, Yifan Zhang, Badrinadh Aila, & Vinod Namboodiri. (2024). NaVIP: An Image-Centric Indoor Navigation Solution for Visually Impaired People.
>
> [InCrowd-VI]Bamdad, M.; Hutter, H.-P.; Darvishy, A. InCrowd-VI: A Realistic Visual–Inertial Dataset for Evaluating Simultaneous Localization and Mapping in Indoor Pedestrian-Rich Spaces for Human Navigation. Sensors 2024, 24, 8164.
>
> [Turn-by-turn Indoor]Santosh Srinivasaiah, Sai Kumar Nekkanti, & Rohith Reddy Nedhunuri. (2024). Turn-by-Turn Indoor Navigation for the Visually Impaired.

---

> > ### Comment · Reviewer_Xgwu · 2025-08-05
> >
> > Thanks to the authors for the detailed and constructive responses. My concerns have been addressed, and I will keep my original rating, recommending acceptance. I believe this dataset will make a valuable contribution to the fields like spatial and embodied intelligence, and encourage the extension to indoor scenes in future work.

---

> > > ### Author Response · Authors · 2025-08-05
> > >
> > > Thank you very much for your feedback. We are glad that our response addressed your concerns.
> > >
> > > Once again, we appreciate your high evaluation of mmWalk and valuable suggestions. We are excited and ready for mentioned future work directions, including extending mmWalk to indoor scenes.
> > >
> > > Sincerely,
> > >
> > > Authors

---

### Note · Authors · 2025-08-12

Dear AC, Reviewers,

We sincerely thank all reviewers for their constructive feedback, thoughtful suggestions, and the recognition of mmWalk’s contributions. Across the **initial reviews**, there was a consistent acknowledgment of the work’s strengths:

- The clear and meaningful motivation in addressing the mobility challenges of blind and low-vision (BLV) individuals, as well as its relevance to spatial and embodied intelligence research.

- The comprehensive and large-scale dataset design, covering diverse perspectives, modalities, scenes, and weather conditions, enriched with BLV-relevant corner cases and panoramic projections.

- The novel multi-modal, multi-view structure and the well-defined mmWalkVQA benchmark, which support fine-grained analysis and extensive evaluation of state-of-the-art VLMs.

During the **rebuttal phase**, we addressed concerns and questions raised by reviewers:

- To the unclarity in the paper, we enhanced explanations of the original content, improving the paper’s readability.

- To address the concern about insufficient experiments, we provided a more detailed description of benchmarking experiment design.

- To alleviate uncertainty raised by LLM evaluation, we supplemented our evaluation with additional human evaluation sampling.

We are encouraged that reviewers explicitly acknowledged our efforts during the rebuttal, informed us that many of their concerns had been addressed, and provided positive evaluations of mmWalk’s contributions. These affirmations reflect the constructive nature of the review process and the alignment between our revisions and the reviewers’ expectations. Moreover, based on the suggestions during **discussion**, we will improve the camera-ready version:

- We will clarify tables, paragraphs, and terms mentioned by reviewers.

- We will present the additional human evaluation sampling results in detail in the appendix.

- We will expand our future work section to include the extension to indoor scenes, enrich the discussion about bridging sim-to-real gap and expand future experiment dimensions.

In summary, we believe the manuscript will become clearer, better-structured, and more comprehensive. We are committed to implementing the above changes in the camera-ready version and to exploring the additional directions suggested by reviewers, ensuring the strength, value and contributions of mmWalk towards BLV assistance, spatial intelligence, and embodied AI research.

Sincerely,

Authors.

---

### Decision · Program_Chairs · 2025-09-18

**Decision:**

Accept (poster)

**Comment:**

This paper introduces mmWalk, a synthetic multi-modal dataset that integrates multi-view sensors and accessibility-oriented features to support safe outdoor navigation, particularly for blind and low-vision (BLV) users. The dataset provides RGB, depth, and semantic segmentation modalities captured from walker, guide dog, and drone viewpoints. In addition, the authors propose mmWalkVQA, a new VQA benchmark built on mmWalk for evaluating vision–language models (VLMs) in assistive tasks. The VQA pairs are generated by GPT-4o, and model responses are compared against GPT-4o’s answers using GPT-4o-mini as the evaluator.

The reviewers agree that the paper is driven by a clear and meaningful motivation. Outdoor navigation for BLV individuals is both a scientifically significant and socially impactful challenge, and the proposed dataset has broader value for advancing research in spatial intelligence. The dataset is large-scale (over 559k panoramic images) and comprehensive, covering multiple modalities (RGB, depth, semantic), three viewpoints (walker, dog, drone), and diverse scenarios and weather conditions, including BLV-specific corner cases. The reviewers also noted that the evaluations include a wide range of state-of-the-art VLMs, and that fine-tuning on mmWalk clearly improves performance on BLV-related tasks.

The major concerns from the reviewers are as follows. First, mmWalk is fully synthetic and restricted to outdoor scenes, with no indoor scenarios. Second, the VQA pairs in mmWalkVQA are generated by ChatGPT-4o, suggesting that GPT-4o may already perform well on this task. Third, the dataset and benchmark have not been validated with human participants from the BLV community; without user studies or practitioner feedback, their practical utility remains speculative. Finally, the novelty is limited, as the work primarily integrates existing tools (CARLA, GPT-4o) rather than introducing new methods.

In the author–reviewer discussion, the authors addressed the reviewers’ concerns with providing additional results including human evaluation. All reviewers recommended acceptance of the paper. The AC concurs with the reviewers’ assessments and therefore also recommends acceptance.